# Glacial sedimentation, fluxes and erosion rates associated with ice retreat in Petermann Fjord and Nares Strait, NW Greenland

Kelly A. Hogan[1,2], Martin Jakobsson[3,4], Larry Mayer[2], Brendan Reilly[5], Anne Jennings[6], Tove Nielsen[7], Katrine J. Andresen[8], Egon Nørmark[8], Katrien A. Heirman[7,9], Elina Kamla[7,10], Kevin Jerram[2], Christian Stranne[3,4], Alan Mix[5]

[1] British Antarctic Survey, Natural Environment Research Council, High Cross, Madingley Road, Cambridge, CB3 0ET, UK
[2] Center for Coastal and Ocean Mapping, University of New Hampshire, Durham, NH 03824, USA
[3] Department of Geological Sciences, Stockholm University, 106 91 Stockholm, Sweden
[4] Bolin Centre for Climate Research, Stockholm University, 106 91 Stockholm, Sweden
[5] College of Earth, Ocean, and Atmospheric Sciences, Oregon State University, Corvallis, OR 97331, USA
[6] Institute of Arctic and Alpine Research, University of Colorado, Boulder, CO 80309-0450, USA
[7] Geological Survey of Denmark and Greenland, Øster Voldgade 10, 1350 Copenhagen K, Denmark
[8] Department of Geoscience, Aarhus University, Hoegh-Guldbergs Gade 2, DK-8000, Aarhus C, Denmark
[9] TNO, Geological Survey of the Netherlands, Princetonlaan 6, NL-3584 CB Utrecht, The Netherlands
[10] Rambøll Management Consulting, Hannemanns Allé 53, DK-2300 Copenhagen S, Denmark

*Correspondence to*: Kelly Hogan (kelgan@bas.ac.uk)

**Abstract.** Petermann Fjord is a deep (>1000 m) fjord that incises the coastline of northwest Greenland and was carved by an expanded Petermann Glacier, one of the six largest outlet glaciers draining the modern Greenland Ice Sheet (GrIS). Between 5-70 m of unconsolidated glacigenic material infills in the fjord and adjacent Nares Strait, deposited as the Petermann and Nares Strait ice streams retreated through the area after the Last Glacial Maximum. We have investigated the deglacial deposits using seismic stratigraphic techniques and have correlated our results with high-resolution bathymetric data and core lithofacies. We identify six seismo-acoustic facies in more than 3500 line-km of sub-bottom and seismic-reflection profiles throughout the fjord, Hall Basin and Kennedy Channel. Seismo-acoustic facies relate to: bedrock or till surfaces (*Facies I*); subglacial deposition (*Facies II*); deposition from meltwater plumes and icebergs in quiescent glacimarine conditions (*Facies III, IV*); deposition at grounded ice margins during stillstands in retreat (grounding-zone wedges; *Facies V*); and the redeposition of material down slopes (*Facies IV*). These sediment units represent the total volume of glacial sediment delivered to the mapped marine environment during retreat. We calculate a glacial sediment flux for the former Petermann Ice Stream as 1080-1420 $m^3$ $a^{-1}$ per meter of ice stream width and an average deglacial erosion rate for the basin of 0.29-0.34 mm $a^{-1}$. Our deglacial erosion rates are consistent with results from Antarctic Peninsula fjord systems but are several times lower than values for other modern GrIS catchments. This difference is attributed to fact that large volumes of surface water do not access the bed in the Petermann system and we conclude that glacial erosion is limited to areas overridden by streaming ice in this large outlet glacier setting. Erosion rates are also presented for two phases of ice retreat and confirm that there is significant

variation in rates over a glacial-deglacial transition. Our new glacial sediment fluxes and erosion rates show that the Petermann Ice Stream was approximately as efficient as the palaeo-Jakobshavn Isbræ at eroding, transporting and delivering sediment to its margin during early deglaciation.

## 1 Introduction

The volume and distribution of glacial sediment in fjords is largely a function of the retreat behaviour of the marine-terminating
glaciers that occupy them. This sediment infill is the final product of material eroded across the catchment area, transported to the ice margin by glacial processes, and ultimately released into fjord basins as grounded ice decays. As a result, these sedimentary archives can provide information about both ice-retreat dynamics, as well as about glacial erosion rates and sediment fluxes, that relate to periods of past climatic warming and associated glacier retreat. The present decay of the Greenland Ice Sheet (GrIS), and its accelerating contribution to sea level rise (Chen et al., 2017), is occurring predominantly
through mass loss from its marine-terminating outlet glaciers (Mouginot et al., 2019). This will likely enhance glacial erosion rates and, therefore, sediment influx to the global ocean. This has implications for marine biogeochemical cycles (e.g., Hawkings et al., 2015) and ultimately organic carbon sequestration in fjord sediments, which are a known "hotspot" for carbon burial (Smith et al., 2015). Yet we have very few estimates of glacial erosion or sediment fluxes for either the present-day GrIS, or for past configurations of the ice sheet (Hallet et al., 1996; Overeem et al., 2017). Obtaining *in situ* observations of
sediment fluxes close to the margins of Greenland's major marine-terminating glaciers remains logistically difficult. Thus, investigations of the sediment infill of fjords provide an important tool for quantifying rates of glacial erosion and sediment transport to the global ocean, as well as for reconstructing glacier behaviour and its drivers beyond the observational record.

Many studies of fjord infill exist for Norwegian, Svalbard and Alaskan fjords; however, in Greenland, ship-based research is hampered by difficult ice conditions and relatively remote locations, issues that generally increase in complexity further
north. Fjords housing major outlet glaciers are often choked by an ice mélange – a dense pack of calved icebergs and sea ice (cf. Amundsen et al., 2010) – that render some fjords almost inaccessible to research vessels. This situation is augmented in northern Greenland by persistent sea-ice cover cementing icebergs together in winter and extending far beyond the coast for up to 11 months of the year (DMI, 2018). As a result, there are only a few previous studies of marine sediments from northern Greenland that are based on sediment cores (e.g., Jennings et al., 2011, 2019; Madaj, 2016; Reilly et al., 2019), and none with
extensive or detailed geophysical mapping of the glacial sediment infill in fjords. For comparison, some geophysical surveys do exist from central and southern Greenland fjords that tend to be ice-free more often (e.g., Uenzelmann-Neben et al., 1991; Andrews et al., 1994; Gilbert et al., 1998, 2002; Ó Cofaigh et al., 2001, 2016; Evans et al., 2002). However, these typically have only a small number of survey lines to represent an entire fjord system, show only a few profiles as exemplars, and/or have not mapped the sediment stratigraphy in detail. This data gap was addressed by the *Petermann 2015 Expedition*, which
collected, in addition to terrestrial, biological and oceanographic datasets (Münchow et al., 2016; Heuzé et al., 2017; Lomac-MacNair et al., 2018), a comprehensive suite of marine geophysical and geological data from Petermann Fjord and the adjacent

part of Nares Strait, northwest Greenland (Fig. 1b, 2) (Jakobsson et al., 2018). Combining systematic classification and mapping of the seismic (acoustic) datasets with seafloor geomorphology provides a means to correlate sediment infill with glacidynamic processes leading to an improved understanding of the Holocene retreat of the Petermann and Nares Strait ice streams (cf. England, 1999; Jakobsson et al., 2018).

This study reconstructs de- and post-glacial sedimentary processes and fluxes in Petermann Fjord and the adjacent stretch of Nares Strait (Fig. 1b) using seismic stratigraphy and seismo-acoustic facies. The objectives are: (1) to map glacial marine sediment units, interpret their seismic stratigraphy, and calculate their volumes; (2) to derive deglacial sediment fluxes and erosion rates; (3) to compare our results with other high-latitude fjord settings (Northern and Southern hemisphere) considering regional variations; and (4) to provide geological boundary conditions for numerical glacier modelling exercises. This is the first study to employ high-resolution seismo-acoustic methods with such a high density of survey lines, meaning that the sediment stratigraphy of a fjord beyond a major Greenland outlet glacier has been revealed and mapped in unprecedented detail. It is also the first dataset from the northern part of the landmass. It sheds new light on glacial erosion rates for Greenland over millennial time-scales, and provides quantitative estimates of the sediment flux to the ocean from a major Greenland glacier.

## 2.1 Environmental setting

Nares Strait is the narrow body of water between northwest Greenland and Ellesmere Island that opens out northwards in to the Lincoln Sea and Arctic Ocean (Fig. 1). The northern part of the strait consists of Robeson Channel, Hall Basin and Kennedy Channel, and is typically around 30 km wide, 400-800 m deep. Kennedy and Robeson Channels have generally smooth seafloors, whereas Hall Basin is somewhat wider (~40-60 km) and has a notably rougher or fractured seafloor beyond the mouth of Petermann Fjord (Jakobsson et al., 2018; Fig. 2). The bedrock geology in the area consists of Precambrian basement rocks capped by Palaeozoic platform limestones that have been dissected by two sets of approximately orthogonal faults trending NNE-SSW and N-S. The Wegener Transform Fault, crossing from Judge Daly Promontory to Kap Lupton (Fig. 2) in the study area and extending northwards in Nares Strait (Dawes 2004; Tessensohn et al., 2006), provides a strong structural control on seafloor morphology in Hall Basin (Jakobsson et al., 2018).

Petermann Fjord is a deep (>1000 m), relatively flat-bottomed fjord with a straight planform shape that is 15-20 km wide. The fjord walls have steep gradients (>70°) resulting in a box-like cross-section. The most prominent bathymetric feature of the fjord is a sill at the fjord-mouth rising to between 350-450 m water depth but with its deepest part (443 m) about 2 km west of the midline of the fjord mouth (Jakobsson et al., 2018). Modified Atlantic Water flows into both Nares Strait and Petermann Fjord from the Lincoln Sea (Münchow et al., 2016; Johnson et al., 2011) but is overlain by a cooler, fresher water mass (Arctic Water) that is also advected into the fjord (Straneo et al., 2012). Oceanographic results from the *Petermann 2015 Expedition* have shown that water in the fjord is dominated by Atlantic Water (AW) at depth (450-600 m) which does not interact with the 40-km long floating ice tongue over the fjord, but is thought to reach the grounding line and contribute to melting there (Münchow et al., 2016; Heuzé et al., 2017). Meltwater from Petermann Glacier was also recorded in all 46 hydrographic casts

collected in 2015 in the fjord and in Nares Strait, with meltwater exiting the fjord on its northern side at water depths of 100-300 m (Heuzé et al., 2017). The present-day retreat of Greenland's marine-terminating glaciers, including Petermann Glacier, has been partly attributed to warming of the AW that reaches the ice margins and enhances frontal melting (Holland et al., 2008; Rignot et al., 2010; Johnson et al., 2011; Straneo et al., 2012; Heuzé et al., 2017; Cai et al., 2017). Furthermore, AW was present in Hall Basin during deglaciation and may have promoted grounded ice retreat during deglaciation (Jennings et

al., 2011).

**2.2 Late Weichselian to Holocene glacial history**

During the Last Glacial Maximum (LGM), the ice sheet in northern Greenland was coalescent with the Innuitian Ice Sheet over Ellesmere Island (England, 1999; England et al., 2006), and grounded ice occupied Nares Strait. The distribution and magnitude of isostatic rebound in the area suggests that ice was at least 1 km thick in the strait, and terrestrial landforms

indicate that Greenland ice extended across to the eastern side of Ellesmere Island (England, 1999). Ice is thought to have been distributed northward and southward from Kane Basin in central Nares Strait (Fig. 1b), with deglaciation of the strait occurring from its northern and southern ends from 11.3 cal. ka BP and 11.7-11.2 cal. ka BP, respectively (recalibrated from England, 1999; Jennings et al., 2019). A sediment core from north-eastern Hall Basin indicates that this area, in front of Petermann Fjord, was free from grounded ice by 9.7 cal. ka BP and was experiencing distal glacimarine conditions by 8.9 cal. ka BP

(Jennings et al., 2011). Further south, dates from a core in Kane Basin show that it had deglaciated around 9.0 cal. ka BP (Georgiadis et al., 2018) (Fig. 1b). Owing to uncertainties in the reservoir corrections for the area and differences in the material dated for deglacial ages, there is still some debate as to when the ice saddle between northwest Greenland and Ellesmere Island disintegrated. However, a recent study by Jennings et al. (2019) discussed this issue in detail and concluded that the strait could have opened as early as 9.0 cal. ka BP or as late as 8.3 cal. ka BP.

Reconstructions of full-glacial ice flow in the area include north-eastward flow out of Nares Strait contributing to eastward flow of ice along the north Greenland coastal plain (Möller et al., 2010; Larsen et al., 2010; Funder et al., 2011). North of Kane Basin, strong convergent flow from the Innuitian and Greenland ice sheets, as evidenced by glacial striae and erratics, probably resulted in an ice stream in Nares Strait (England et al., 2006). This flow pattern is supported by recent mapping of submarine landforms including mega-scale glacial lineations (MSGL) in Kennedy and Robeson Channels which indicate northward

movement of fast-flowing grounded ice in the strait, most likely representing the late deglacial imprint of grounded ice activity (Jakobsson et al., 2018). A change in lineation orientation close to the mouth of Petermann Fjord was interpreted as a signature of ice exiting the fjord and merging with ice flow in Nares Strait causing a slight deflection in the flow pattern (Jakobsson et al., 2018).

By combining terrestrial evidence with the submarine landform record, Jakobsson et al. (2018) suggested the following

sequence of events for the deglaciation of northern Nares Strait and Petermann Fjord (see Fig. 1b for location). All ages were inferred by correlating the mapped marine landforms to dated ice margins on land by England (1999). Since the ice margins

on land were presented as uncalibrated $^{14}C$ years BP (England, 1999), calibration to calendar years was made by Jakobsson et al. (2018) using the Marine13 radiocarbon age calibration curve (Reimer et al., 2013) and a $\Delta R = 268 \pm 82$ years.

At 9.3 cal. ka BP ($1\sigma$ range: 9440-9140 cal. a BP) the retreating ice margin was grounded between Kap Lupton and the Judge Daly Promontory (Fig. 1b) along a prominent bathymetric shoaling (S4 on Fig. 2). At this time, there is evidence for abundant meltwater release and ice stagnation on the eastern side of Hall Basin. By 8.7 cal. ka BP ($1\sigma$ range: 8835-8459 cal. a BP) the ice margin is thought to have retreated to the mouth of Petermann Fjord where it rested on the prominent fjord-mouth sill (Fig. 1b) and was probably fronted by an ice tongue. A significant sedimentary wedge – a grounding-zone wedge (GZW) – built up on the sill and reinforced ice-margin stability at this location (cf. Alley et al., 2007; Dowdeswell and Fugelli, 2012). The ice margin subsequently lost its ice shelf and retreated down the backside of the sill as a tidewater glacier cliff, possibly due to catastrophic calving by a process termed marine ice cliff instability (Pollard et al., 2015). Based on terrestrial dates this is inferred to have occurred around 7.6 cal. ka BP ($1\sigma$ range: 7740-7495 cal. a BP), after which the retreat of grounded ice through the remainder of the fjord was rapid. Recent sedimentological work suggests that the fjord was probably not covered by a floating ice tongue directly after this rapid retreat of the grounded Peterman Ice Stream (which became Petermann Glacier), for around 5000 years in the mid-Holocene (Reilly et al., 2019). The modern glaciologic setting, which includes a 40-km long floating tongue, did not develop until c. 2.2 cal. ka BP (Reilly et al., 2019).

# 3 Data and methods

## 3.1 Geophysical datasets

Two primary geophysical datasets were used in this study: high-resolution, sub-bottom profiles (SBP) and airgun seismic-reflection profiles (AG), both collected during the *Petermann 2015 Expedition* to the Petermann Fjord and Nares Strait area in 2015 on the Icebreaker (IB) *Oden*. More than 3100 line-km of SBP were acquired using the hull-mounted parametric Kongsberg SBP 120, which transmits a low-frequency (2.5-7 kHz) chirp pulse with a narrow (3°) main beam. Vertical resolution of the SBP profiles is approximately 0.35 ms (~70 cm using a sediment velocity of 1500 m s$^{-1}$). Penetration was up to 60 m in unlithified sediments and the quality of the SBP data was generally good, although frequently influenced by noise from ice breaking. Two artefacts are prominent in the data: (i) on steep slopes, side echoes and the scattering of acoustic energy resulted in returned reflections being diffuse, and (ii) a rugged and hard seafloor generated numerous sidewall echoes and hyperbolae. Line spacing was generally as low as 600 m and rarely exceeded 2.5 km. The multidisciplinary nature of the expedition required an abundance of sampling stations and, in turn, resulted in numerous crossing lines and multiple transects of key areas (Fig. 2). The nature of deeper sediments and bedrock structure were studied using 10 AG profiles (Fig. 2) acquired with a single airgun source (210 cu. in. Generator Injection (GI) gun with a firing interval of 5 s and a record length of 3 s). The streamer had a total active length of 300 m with 48 hydrophone groups (8 hydrophones each) and was towed at depths of 7-16 m. Navigation for the SBP profiles was taken directly from the ship's Seatex Seapath 320 GPS feed. Motion correction of the SBP data was applied using information provided by the installed Seatex MRU5 motion reference unit. For the AG

profiles, a separate Thales DG16 GPS system was used to calculate positions and offset geometries for the ship, source, and

hydrophones. Heritage seismic-reflection profiles acquired in 2001 were also available and were used to investigate the character of key glacial landforms. These data were acquired by Bundesanstalt für Geowissenschaften und Rohstoffe (BGR) using 6 GI guns and a 48-channel array (24 hydrophones each) in a 100 m-long streamer (shortened due to ice conditions). Details of the acquisition and processing of this dataset (2001 BGR lines on Fig. 2) are provided in Jackson et al. (2006).

       Processing of the SBP data involved calculation of instantaneous amplitudes from the correlated SBP 120 output which

were then visualized as variable density traces in open-source software (dgB Earth Sciences OpendTect v6.4.0). The 2015 AG profiles were processed using standard processing techniques including geometry definition, amplitude correction and bandpass filtering preserving data in the frequency range of 40-350 Hz. FK-filtering was applied in order to remove propeller noise. After CDP (common depth point) stacking and migration, a gentle trace mix and automatic scaling was also applied. The output AG data were interpreted in Petrel 2015 and OpendTect. The seismic datasets were analysed alongside a gridded

3D surface of the seafloor produced from high-resolution bathymetric data also acquired during the *Petermann 2015 Expedition*. The bathymetric data were collected using a Kongsberg EM122 (12 kHz) multibeam echosounder with a 1° (TX) x 1° (RX) array. Data coverage and water depths in the area resulted in the final grid having 15-m square grid cells. Detailed information and interpretation of the multibeam bathymetric dataset is presented in Jakobsson et al. (2018).

**3.2 Seismic data interpretation**

All output profiles are in two-way travel time (TWT). Seismo-acoustic facies were identified primarily from SBP profiles based on reflection geometry, reflection strength, and continuity; these were cross-checked on AG profiles and one additional facies (IV) was identified only on AG profiles. For the SBP data, the profiles were inspected and a coherent and continuous, high-amplitude reflection (R1) at the base of the uppermost unlithified sediment package (often marking the acoustic basement) was identified and digitized (Figs. 3a, c). In general, this reflection was picked manually because auto-tracking methods in

OpendTect could not be used due to the variable penetration of the SBP 120, the rugged nature of the reflection, the noise artefacts noted above, and some limitations with the 2D picking algorithm. R1 picks on SBP profiles were supplemented and verified by the deeper-penetrating AG lines (Figs. 3b, d). R1 picks were gridded to make separate 3D surfaces for Petermann Fjord and Nares Strait, based on the separation of these areas by the shallow sill at the fjord-mouth (over which the unlithified sediments disappear on SBP profiles) and the known glacial history of the area (see Section 2.2). Isopach maps for the

unlithified sediment package were produced by subtracting the depth-converted R1 surface from the multibeam bathymetric digital elevation model of the seafloor; in general, stratigraphic thicknesses in metres in this study have been calculated using a sediment sound velocity of 1500 m s$^{-1}$ (cf. Nygård et al., 2007; Hjelstuen et al., 2009; Hogan et al., 2012). Key glacial landforms, predominantly GZWs in this case, were also identified and mapped on the AG and SBP profiles because these landforms indicate former grounding-line locations. Significantly, when they are tied with deglacial chronology, these

landforms can be used to calculate sedimentary processes associated with specific time periods and stillstand events (e.g., Callard et al., 2018; Nielsen and Rasmussen, 2018). Base GZW reflections were digitized where AG profiles exist over these

features and where they were visible on SBP profiles. These were gridded using the 'surface' splines in tension algorithm in GMT (Smith and Wessel, 1990), converted to depth below the seafloor, and used to calculate GZW volumes. For the GZWs, volumes were calculated with sediment velocities of 1500 m s$^{-1}$ but also with the higher value of 1800 m s$^{-1}$. The latter value is based on previous estimates of velocities in subglacial tills from (over-ice) seismic data (e.g., Smith, 1997; Tulaczyk et al., 1998; King et al., 2004), including recent measurements from Greenland (Hofstede et al., 2018) and on the measured physical properties of coarse shelf sediments including diamictons (e.g., Hamilton, 1969; Cochrane et al., 1995). Thus, for GZW thicknesses and volumes a range of values is given.

Three sediment cores acquired during the *Petermann 2015 Expedition* and one previously published core (Table 1; locations in Fig. 2) were used to correlate seismo-acoustic facies with lithofacies. The correlation was based on comparing SBP profiles at individual core sites with core CT-scans, photographs and descriptions. Full descriptions of the methods used to acquire the cores and the CT-scans can be found in Reilly et al. (2017, 2019).

### 3.4 Glacial volumes, fluxes and erosion rates

In high-latitude fjords and glacial troughs beyond the coastline, the unlithified sediment accumulation may be taken to represent material deposited since these areas were last occupied by grounded ice, during ice retreat following the LGM (e.g., Aarseth, 1997; Gilbert et al., 1998; Hjelstuen et al., 2009; Hogan et al., 2012; Bellwald et al., 2016; Callard et al., 2018; Neilsen and Rasmussen, 2018). This glacial-marine sedimentation has two components (Fig. 4). The first component is coarse or mixed material delivered to the grounding zone subglacially but deposited seaward of there by gravity-flow processes (dark grey on Fig. 4); the second is predominantly fine-grained units (with some coarser particles) that settle out from meltwater plumes within several tens of kilometres from the grounding line ("plumites"; Hesse et al., 1997; yellow on Fig. 4) and as ice-rafted debris (IRD). Here, we have mapped total glacial-marine sediment volumes for the Petermann Fjord-Nares Strait system and then used these volumes to calculate glacial sediment fluxes and erosion rates.

We make two adjustments to the mapped volumes before calculating sediment fluxes and erosion rates. First, we assume that our marine study areas were fully excavated (to bedrock) by grounded ice during the previous glacial event. This assumption is required because we are not able to distinguish pre-LGM sediments in our data even though we acknowledge that older (pre-LGM) sediment is sometimes preserved in these settings (e.g., Hooke and Elverhøi, 1996; Aarseth, 1997; O'Regan et al., 2017; Jennings et al., 2019). This assumption is justified by theoretical studies of glacial erosion/sediment transport, which are based on observations, that most often suggest that fjords are rapidly and fully excavated during glacial advances (Powell, 1984; Aarseth, 1997; Hjelstuen et al., 2009). Likewise, this assumption is usually applied to studies of fjord sediment volumes used to calculate glacial erosion rates (e.g., Powell, 1991; Hunter, 1994; Hallet et al., 1996). We derive some support for this assumption from the seafloor morphology of Petermann Fjord and Hall Basin. Ice-sculpted bedrock surfaces of probably LGM age are clearly visible across much of the area (Jakobsson et al., 2018), indicating that significant pre-LGM sediments most likely do not remain in the fjord. We also note that, although it is possible for pre-LGM sediments to be present, if they are too deep to be resolved on our SBP profiles then they have not been included in our volume estimates.

Furthermore, if they are preserved as the lowermost part of the stratigraphy (i.e., most likely in basins), they are probably not volumetrically significant because there are relatively few basins containing thick sediments in the area (e.g., Fig. 12). Still, we acknowledge this potential source of error in our estimations. Second, we elect to remove 0.5 m of sediment cover for all mapped areas because dates from nearby sediment cores reveal that the upper ~0.5 m of the stratigraphy was deposited after the ice margin had retreated into the fjord (Jennings et al., 2011, 2018); essentially, these uppermost sediments are not

associated with a nearby ice margin. We also assume that other sediment sources (biogenic, aeolian, sidewall erosion) are volumetrically insignificant, which is typically the case in polar fjord settings (Powell, 2005). This is supported by total organic carbon (TOC) measurements on core tops from the area that return extremely low percentages TOC (<<0.5 %) (Jennings, *pers. comm.*), and by the lack of widespread flow deposits in the fjord or in Hall Basin (e.g., Figs. 7-9).

Ultimately, glacial sediment fluxes are calculated simply by dividing the total glacial sediment volume by the time that

the ice margin was supplying sediment to the area, which is taken from existing deglacial chronologies. In order to compare to previous estimates of glacial sediment flux, this number is further divided by the ice stream width to return a sediment flux per ice stream width (cf. Alley et al., 1989). Grounding-line lengths, which is the same as the ice stream width, were measured along the fronts of the Petermann GZW from the multibeam bathymetric surface. Chronological information follows the deglacial reconstructions of England (1999) and Jakobsson et al. (2018); 1σ uncertainties in the ages for the Jakobsson et al.

(2018) ice margin positions are used to provide an error bar on the calculated fluxes (see Section 4.5).

Glacial erosion rates were calculated by applying the methodology outlined in Fernandez et al. (2016) to our volumetric results to calculate the average basin- and time-averaged erosion rates ($\bar{E}$) through:

$$\bar{E} = Vol_{Rx} / (A_{dr} \text{ x } T) \tag{1}$$


where $Vol_{Rx}$ is the volume of (dry) rock, $A_{dr}$ is the effective drainage basin area (in km$^2$) and $T$ is the time for sediment accumulation (in years), in effect the time since ice was near enough to supply sediment to the area concerned. Erosion rates are calculated for the erosion of lithified rock and, therefore, our wet sediment volumes had to be converted to dry rock volumes ($Vol_{Rx}$). This was done using a wet density ($\rho_{sed}$) of 1850 kg m$^{-3}$ for the sediments (based on measured density values from

*Petermann 2015 Expedition* cores) and a density ($\rho_{source}$) of 2700 kg m$^{-3}$ for the source rocks (a commonly used density for parental rock types gneiss and limestones, following Andrews et al., 1994 and Fernandez et al., 2016).

## 4 Results and interpretation

### 4.1 Seismo-acoustic facies and depositional environments in Petermann Fjord and Nares Strait

We identify six seismo-acoustic facies in Petermann Fjord and the adjacent area of Nares Strait (Fig. 5) and correlate these

with core lithologies where possible. Seismo-acoustic facies I (bedrock), II (subglacial till) and VI (GZW units), as defined below, were not recovered by either piston or gravity cores, generally because they were too deep in the stratigraphy to be

sampled or are not able to be sampled by these types of coring devices (i.e., bedrock). Therefore, only seismo-acoustic facies III, IV and V were correlated with sediment lithofacies (Fig. 6). The seismo-acoustic facies are:

(I) *Acoustically-impenetrable to homogenous facies.* This facies is represented by a high-amplitude, prolonged reflection defining a rugged surface with rare sub-bottom point and diffraction hyperbolae on slopes. It marks the base of the acoustic stratigraphy on SBP profiles and we interpret it to be bedrock or a till surface. The SBP data alone do not allow us to differentiate between these two types, but by correlating with AG lines where seismic basement is reached we can identify this facies as bedrock in Hall Basin. However, in areas where glacial lineations (which are formed subglacially in deforming till) are present, the upper reflection of this unit is interpreted to be a till surface (e.g., Fig. 7b).

(II) *Acoustically-homogenous, non-conformable facies.* This unit has a strong, prolonged upper reflection and a lower amplitude basal reflection that can be discontinuous. It is acoustically-homogenous and shows varying thickness that is not conformable with the basal reflection or underlying units. In areas where this unit is correlated with MSGL it is interpreted as a subglacial till layer (cf. Ó Cofaigh et al., 2005); where this unit occurs on seafloor highs in Nares Strait it is interpreted as an iceberg ploughed or current-reworked facies based on correlation with iceberg ploughmarks on the multibeam-bathymetric data.

(III) *Acoustically-stratified, conformable facies.* This is characterized by parallel to sub-parallel, continuous, high- to medium-amplitude reflections with conformable geometries. It is typically 5-15 m thick. We interpret this facies as glacimarine and/or hemipelagic sediments primarily deposited via suspension settling (with variable IRD) in an ice-distal setting. Support for this interpretation comes from lithofacies correlation with the upper part of core OD1507-41GC from Petermann Fjord and core HLY03-05GC which samples the facies from Robeson Channel just beyond the S4 ridge (Fig. 2). The upper part of core OD1507-41GC recovered a brown, homogeneous clay with dispersed sand and clasts (Figs. 6a, e) consistent with distal glacimarine sediments deposited largely from suspension with dispersed IRD (cf. Elverhøi et al., 1989; Powell & Domack, 2002) and also described in cores from the fjord by Reilly et al. (2019). The sedimentology of HLY03-05GC was described by Jennings et al. (2011) who identified bioturbated muds to 112 cm, a transitional laminated pebbly mud unit from 112-125 cm and a laminated mud unit from 125 cm to the base of the core (Fig. 6b, f). The laminated units were interpreted as distal glacimarine sediments with the transitional unit reflecting the breakup of ice in Nares Strait and Kennedy Channel (Jennings et al., 2011).

(IV) *Acoustically-stratified basin or onlapping fill.* This facies also comprises parallel to sub-parallel, continuous, high- to medium-amplitude reflections either in a ponded basin-fill geometry (reflections terminate at basin sides) or in an onlapping fill geometry (reflections curve up the flanks of basins). It can include acoustically-transparent bodies, usually several meters thick that pinch out laterally. This facies is interpreted as a combination of suspension settling of glacimarine and hemipelagic sediments and gravity-flow deposits (GFDs) forming the acoustically-transparent bodies (see *Facies V*) made up of material redeposited into basins from nearby slopes. Core OD1507-37PC penetrated *Facies IV* (Fig. 6g) in Petermann Fjord in a basin around 2 km from the ice-tongue margin as it was in 2015 (Fig. 2). It sampled grey-brown clay with dispersed clasts interrupted by multiple coarse, sand units (typically < 10 cm thick) that are normally-graded and have sharp basal contacts (Fig. 6c). The

clay with clasts is interpreted as glacimarine sediments with IRD and the individual sand units have properties consistent to gravity-flow deposits (GFDs) (i.e., erosive at their base and grading upwards; e.g., Bøe et al., 2004; Gilbert et al., 2002). Although such thin sand units may not be resolved in the acoustic SBP data over the core site, this lithofacies supports our interpretation of the seismo-acoustic facies as glacimarine units with interbedded GFDs.

(V) *Acoustically-transparent facies.* Multiple reflectors in Kennedy Channel comprise this facies in lens-shaped or tapered bodies on slopes. This facies is also present in local basins where it often pinches out towards the basin flanks, both in Petermann Fjord and Hall Basin. The lensoid and pinching-out geometries of these units, their erosion of underlying sediments, and their acoustically-transparent nature are characteristic of GFDs (cf. Laberg and Vorren, 2000; Hjelstuen et al., 2009). Core OD1507-52PC (Fig. 6d) was recovered from a stratigraphy that included discrete lenticular bodies (Fig. 6h) and it sampled

laminated muds interbedded with diamictic units with sharp contacts consistent with sediment flow deposits consisting of glacigenic debris (cf. Laberg & Vorren, 2000). Note that the individual diamictic units in the core are not resolved in the SBP data as discrete reflections.

      (VI) *Downlapping to chaotic facies.* This facies is only seen on the AG profiles over the GZWs in the area. It consists of low-amplitude chaotic point reflections and rare discontinuous, sub-parallel reflections forming either a layered or

downlapping pattern. The location of this facies at a known GZW location (Jakobsson et al., 2018), and its seismic character, are consistent with its interpretation as subglacial till forming a GZW on a bathymetric high (e.g., Anderson, 1999; Dowdeswell and Fugelli, 2012). Deposition most likely occurred via subglacial plastering (aggradation) on the ice-proximal slope of the wedge and proglacially (i.e., seaward of the grounding line) via small gravity flows on the ice-distal slope (progradation). These processes probably occurred asynchronously with aggradation during advance of the grounding line over the sill and

progradation only occurring when the grounding zone was on the sill. Thus, the GZW on the sill may be more of a combined morainal bank with GZW on its upper part, rather than a wedge-shaped GZW in its traditional form.

## 4.2 Glacial marine sediment infill in Petermann Fjord and Nares Strait

The deepest part of Petermann Fjord, lying inside of (SE of) the mouth sill within the steep sidewalls (up to 70° slopes), is generally draped by a 5-15 m of *Facies III* (Figs. 5, 7). This unit conformably overlies the rugged surface of *Facies I* (Fig. 5).

The seafloor morphology of the fjord bottom, which comprises relatively flat-lying parts separated by steep "steps" and has been strongly sculpted by ice (Jakobsson et al., 2018), suggests that the basal reflection here usually represents bedrock.  In the few small areas where glacial lineations have been identified (e.g., around 61° 39'W, 81° 03.5' N), the basal reflection on SBP profiles represents a subglacial till surface (Fig. 7b). On the terraces on the western side of the fjord, bedrock is covered by about 5 m of draped *Facies III* overlying a thin unit of *Facies II*; these are interpreted as glacimarine/hemipelagic sediments

overlying a plastered till unit.

      On the eastern side of the fjord and in some places in the mid-fjord area, about 25 km from the 2015 ice tongue margin, local basins in the bedrock surface are filled with at least 35 m of stratified sediments (Fig. 7c). This basin fill is typically ponded in basins in the central part of the fjord, and has an onlapping geometry with relatively more transparent sub-units in

basins on the eastern side of the fjord (*Facies IV*; Fig. 6g). We interpret these both as glacimarine/hemipelagic sediments with the onlapping fill including interbedded GFDs promoted by increased sediment input from two small glaciers entering the fjord there (Belgrade Glacier and Unnamed Glacier; Fig. 2). Some basins in the central fjord also contain sediment gravity flow deposits (Fig. 5b) presumably representing material redeposited from local slopes. From the seafloor morphology, we note that there are two clear fan-shaped deposits in the fjord immediately seaward of the margins of Belgrade and Unnamed glaciers, which are interpreted as ice-proximal fans (e.g., Fig. 12). Unfortunately, the SBP profiles do not penetrate the fan deposits and we do not have AG profiles in this area.

In the Hall Basin area of Nares Strait, between the Petermann fjord-mouth sill and the S2 high (Fig. 2), the seafloor deepens to 500-620 m and includes several small basins (1 to >10 km$^2$), sometimes interconnected and expressed as flat areas of seafloor interrupted by rugged seafloor highs. The highs comprise *Facies I*, are variously ice sculpted (Jakobsson et al., 2018), and are easily interpreted as bedrock. In the basins, the unlithified sediment package consists of stratified basin fill with GFDs (*Facies IV*) up to 45 m thick. Between basins, bedrock is mantled by 10-15 m of *Facies III*. Together these units are interpreted to be the product of rainout of glacimarine and hemipelagic material that forms conformable layers over bedrock where slopes are relatively gentle, but is focused into basins by redeposition from nearby slopes (gradients up to 20°). The largest flow deposits (GFDs) are apparent as thick (>10 m) acoustically-transparent bodies (Fig. 8), and indicate that redeposition from the basin sides is an important process locally. They are correlated with the flattest basin floors with sharp, well-defined basin edges showing that sediment has run in to the basin and then been dammed by a bedrock high (Fig. 8a). On the most prominent bedrock highs (S2-S4; Fig. 2), unlithified sediments consist of *Facies II* and are usually <8 m thick (Fig. 8b). However, in deeper areas (> 350 m water depth) the rugged bedrock surface is mantled with 7-15 m of *Facies III* (Fig. 8). We interpret this pattern to reflect a dominance of rainout processes that uniformly draped bedrock/till with up to 15 m of layered sediments unless: (i) material was redeposited down-slope and into basins, or (ii) strong currents in Nares Strait (e.g., Mudie et al., 2006; Münchow et al., 2006) prevented the deposition of fine-grained material on the highest seafloor areas. Iceberg ploughing also probably helped to homogenize sediment layers deposited on the highs (cf. iceberg ploughmarks on S2 in Jakobsson et al., 2018).

In the >500 m deep and relatively flat Kennedy and Robeson Channels (Figs. 1, 2), unconsolidated sediment comprises a two-layer conformable stratigraphy of *Facies III*. The upper unit is acoustically-stratified and is typically 5-10 m thick. The lower unit, which is separated from the upper unit by a high-amplitude reflection (R1 in Fig. 3), is also 5-10 m thick and conformable but can be either acoustically homogenous or acoustically stratified (e.g., *Facies III* on Fig. 5). Where the bottom unit is homogenous on SBP profiles it has a stratified character on AG lines (cf. Fig. 3). We interpret this as reflection of the SBP acoustic signal at R1 and, therefore, poor penetration of acoustic energy in to the bottom unit. MSGL in Kennedy Channel are formed in *Facies II* interpreted as a subglacial till. A similar interpretation is made for MSGL in Robeson Channel where the MSGL are also formed in *Facies II* but underlie 5-10 m of *Facies III* as described above.

## 4.3 Large subglacial landforms: grounding-zone wedges (GZWs)

There are two large discrete sedimentary deposits in the study area that must be accounted for here because they represent direct glacial sediment delivery to the grounding line and so should be included in any glacial flux and erosion calculations. These are the two GZWs in the study area, one on the Petermann fjord-mouth sill that was identified by Jakobsson et al. (2018) and one in Kennedy Channel around 64° 39' W, 81° 09' N identified in this work from the SBP data (Figs. 9, 10). Both of these features are well covered by SBP lines and the Petermann GZW is also crossed by four AG profiles.

SBP profiles across the Petermann GZW show very limited penetration through this deposit. It has a high-amplitude reflection at its top and is otherwise acoustically-impenetrable (*Facies I*). Only small mounds of acoustically-homogenous material occur above this reflection; these were interpreted as recessional moraines based on their coincidence with small, sinuous ridges in the multibeam dataset (Supp. Fig. 3 in Jakobsson et al., 2018). AG profiles over the GZW provide some more information about its internal character (Fig. 9c). The GZW appears to contain several conformable reflections in its upper 50 ms (~37-45 m) that down-lap at the base of the slope (Figs. 5, 9c, d); however, the reflections have low amplitudes and are discontinuous. Below these reflections the seismic character is poorly defined and chaotic (*Facies VI*) presumably because the deposit consists of a similar lithology throughout and, therefore, contains few acoustic impedance contrasts. However, the base of the deposit can be mapped along about 50 % of its length (e.g., Fig. 9c, d) and defines a surprisingly thick deposit (200 ms TWT; ~150-180 m) that is continuous down the back slope of the fjord-mouth sill. We interpret this seismo-acoustic facies to be a diamictic deposit probably consisting of subglacial till plastered on to the sill by a formerly-expanded Petermann Glacier. Coarse grains in the till deposit result in strong scattering of acoustic energy, making this deposit effectively impenetrable with the SBP source. It is notable that the GZW does not appear to contain the prograding reflections described from some GZWs (e.g., Larter and Vanneste, 1995; Anderson, 1999; Dowdeswell and Fugelli, 2012); we attribute this to its position on the back-slope and upper ridge of the fjord-mouth sill. In this setting, it is difficult to see how a wedge would be built up by progradation up a slope (i.e., on the back-slope of the sill). The deposit has instead been built by plastering of layers of material on the back-slope and possibly through progradation on the top of the sill.

The Kennedy Channel GZW has a different geometry, position and architecture (Fig. 10). The GZW rises 10-15 m from the surrounding seafloor, is at least 5 km wide (along Kennedy Channel) and 7 km long (across Kennedy Channel). Although the multibeam echosounder coverage extends only to the mid-line of the strait, we note that the bathymetry shallows towards Ellesmere Island in this area (based on our multibeam dataset and IBCAO regional bathymetry; Jakobsson et al., 2012), meaning that the GZW persists across the deepest part of the strait. It has a convex-up expression in the bathymetry that is clearly marked by iceberg ploughmarks (Fig. 10a) and is situated in current water depths of ~450 m just south of a marked slope to deeper waters (~530 m) to the north (Fig. 10a). SBP profiles reveal that the deposit comprises 1-3 acoustically-transparent units with variable thicknesses demarked by weak sub-bottom reflections (*Facies V*; Figs 10c, d). AG lines in this area, which do not extend across the mapped GZW and do not fully image the deposit (Fig. 10a), reveal a chaotic seismic character (*Facies VI*) sometimes forming lenticular bodies. However, the deposit thins and eventually pinches-out to the north

(Figs 10c, d). We interpret this acoustic signature as layers of till delivered to the grounding line and then deposited just
seaward of it (probably by gravity flows) at the temporarily stabilized grounding zone of the Nares Strait Ice Stream. The ice
margin stabilized at a bathymetric shallowing and narrowing of the deepest channel in this area. Subglacial till extruded from
the grounding line as GFDs formed the acoustically-homogenous units (*Facies V*) extending and tapering down-slope in front
of the GZW (Fig. 10d). Where such flow deposits are prolific and occur at the seafloor, they are easily identified as smooth,
lobate features in front of known grounding-zone positions marked by terminal moraines (e.g., Ottesen and Dowdeswell, 2006;
Flink et al., 2015) or GZWs (e.g., Bjarnardóttir et al., 2013; Esteves et al., 2017). Here, they may reflect local shifts in the
location of the grounding zone during a phase of ice-shelf instability interpreted from core records (Jennings et al., 2018) prior
to further grounding-zone retreat.

### 4.4 Unlithified sediment volumes

Total sediment thicknesses (to acoustic basement) were mapped from SBP profiles in two areas: Petermann Fjord and inner
Hall Basin (Fig. 11). The isopach map for Petermann Fjord indicates that sediment thicknesses, typically 20-40 m, are relatively
consistent on the fjord bottom with a few depressions holding 70 m of sediment (Fig. 11a). The total mapped sediment volume
in the fjord was 14.2 km$^3$. In Hall Basin, mapping was confined to the area in front of the Petermann sill and south of ridges
S1-S3. This was primarily because the sill is a known grounding-zone location during ice retreat (Jakobsson et al., 2018) and
because that area contains the majority of the sediment-filled basins in front of the sill and up to the topographic barrier at S2-
S4. Secondary to this, the area beyond the S1-S3 ridges has a heavily fractured morphology with many small, isolated basins
and trenches; these features complicate calculations of sediment thickness when survey lines have irregular spacing that is
often greater than the distance between individual basins. However, mapping and the resultant isopach map for this area
indicates sediment thicknesses are typically less than 30 m but up to 50 m in basins, which become more irregular in shape
further northwards (Fig. 11b). The strong correlation of sediment thickness with seafloor morphology confirms that topography
is a strong control on accumulation in this area. The total mapped sediment volume between the fjord-mouth and the S1-S3
ridges is 16.3 km$^3$ (using a sound velocity of 1500 m s$^{-1}$).

The isopach map for the Petermann GZW shows a maximum sediment thickness of 215-260 m on the upper part of the
back-slope of the fjord-mouth sill (Fig. 9b). The thickest part of the deposit appears to be confined to a central fjord-parallel
line, which is likely a function of gridding from a single line in the central part of the fjord (line 04a; Fig. 9a) and probably
leads to an underestimation in sediment thicknesses for the GZW. However, a second line across the southern part of the sill
(line 13b) confirms that the GZW does not extend off the top part of the sill in this area (Fig. 9b). Sediment thicknesses on the
top of the sill are generally between 30-120 m and reach 160-190 m in its northern part. The shape of the GZW is defined by
a zero-thickness contour as mapped on AG profiles joined by tracing along the front scarp of the wedge and extending down
the deepest channel into Petermann Fjord. A volume calculation for the isopach map representing the GZW at mouth of the
Petermann Fjord gives a total volume of 7.7-15.1 km$^3$ (using sound speeds of 1500 and 1800 m s$^{-1}$). In Kennedy Channel, AG
profiles do not fully cover the GZW (Fig. 10a); however, its volume has been estimated based on AG profiles and from SBP

profiles that image the base of the deposit near its edges. The deposit here is more classically wedge-shaped (cf. Alley et al., 1989; Dowdeswell and Fugelli, 2012) compared to the deposits of the GZW at the Petermann Fjord mouth. The Kennedy Channel wedge has the greatest thickness toward the centre of the channel in the frontal part of the wedge (65-78 m; Fig. 10b). The total sediment volume for the mapped part of the wedge is 1.1-2.2 km$^3$; however, our data covers only about half the width of the channel and we recognize that the deposit could be larger.

## 4.5 Glacial fluxes and erosion rates

Using our mapped volumes we can calculate glacial sediment fluxes and erosion rates for the palaeo Petermann Ice Stream during its retreat from the fjord mouth. By adding the Petermann GZW volume (7.7-15.1 km$^3$) to the volume of unlithified sediments in inner Hall Basin (16.3 km$^3$), we obtain a total glacial sediment volume of 24-31.4 km$^3$ that was delivered by the Petermann Ice Stream during deglaciation, when it was located at the fjord mouth. It is not yet known whether the GZW was produced over multiple glacial cycles so we assume, for the purposes of these calculations, that the entire GZW was deposited during the last glacial period. The result is a total sediment volume of 23.8-31.2 km$^3$ (when adjusted to remove the upper 0.5 m of non-glacial sediment). If this volume was deposited over the ~1100 years when the ice margin was stable at the fjord mouth (England, 1999; Jakobsson et al., 2018), it indicates a glacial sediment flux for the Petermann Ice Stream of 1080-1420 m$^3$ a$^{-1}$ m$^{-1}$. Using the 1σ uncertainties in ages for the Jakobsson et al. (2018) ice margin positions (maximum time at the fjord mouth 1340 yrs; minimum time at the fjord mouth 720 yrs) we can give the associated uncertainty in these fluxes as 890-2170 m$^3$ a$^{-1}$ m$^{-1}$. However, we acknowledge the remaining uncertainties with these estimates due to the possibility that some material from the GZW was produced by a previous glacial event and also that some sediment may bypass the system (Petermann Fjord and Hall Basin) in icebergs that melt out elsewhere; it is not possible to quantify these volumes based on currently available data.

To calculate an average deglacial erosion rate for the palaeo Petermann Ice Stream, we must add the volumes of Hall Basin and GZW sediment (23.8-31.2 km$^3$) to the unconsolidated sediment fill in the fjord (14.2 km$^3$). Converting this total volume (38.0-45.4 km$^3$) to its rock-equivalent volume returns a dry volume of 26.0-31.1 km$^3$. Using an effective drainage basin area ($A_{dr}$) of 10 493 km$^2$ for the Petermann Ice Stream (see Supplementary Information and Fig. S1 for drainage basin definitions) and a sediment accumulation time (effectively the time since grounded ice had retreated from the fjord-mouth sill) of 8700 years (after Jakobsson et al., 2018), the average deglacial erosion rate is calculated as 0.29-0.34 mm a$^{-1}$. For the palaeo-Petermann catchment we note that its area could not be significantly larger than the modern drainage basin because the ice stream was constrained to the fjord during deglaciation and the grounding line was at the fjord-mouth sill. Thus, we simply add the deglaciated area of the fjord to the modern Petermann catchment where ice velocities are high enough to allow glacial erosion and transport (i.e., where ice is at the pressure melting point and is not frozen to the bed). For this estimate, we have taken this as the area with (modern) ice velocities >50 m a$^{-1}$ from the MEaSUREs v2 dataset, 2017-2018 velocities (Howat, 2017) (Fig. S1a).

Because of the physiography of the Petermann Fjord system, we are able distinguish between an "earlier" deglacial sediment volume (Petermann GZW and Hall Basin units), when the grounding line was on the fjord-mouth sill and deposition was only on the sill and in Hall Basin, from a "later" deglacial sediment volume (Petermann Fjord units) when grounded ice was retreating through the fjord. Using chronologies from Jakobsson et al. (2018), we can estimate an erosion rate for these two phases of deglaciation. Calculated $\bar{E}$ for 8.7-7.6 ka when the Petermann Ice Stream was at the fjord mouth is 1.41-1.85 mm a$^{-1}$. For the later phase, recent core chronologies show that the fjord was covered by a floating ice tongue by 6.9 ka (Reilly et al., 2019), and therefore must have been free from grounded ice by that time. This implies grounding-line retreat through the fjord in as little as 700 years. Assuming, as before, that all but the upper 0.5 m of fjord infill was deposited during this retreat returns a second-phase deglacial $\bar{E}$ (7.6 ka to present) of 0.14 mm a$^{-1}$.

One outstanding issue with this method of calculating glacial erosion rates is the potential storage of glacially-derived material elsewhere in the system (cf. Cowton et al., 2012; Fernandez et al., 2016). Based on cores recovered from beneath the floating Petermann Ice Tongue (Reilly et al., 2019) there is at least some unconsolidated sediment cover beneath the tongue, and the modelled bathymetry there (based on a gravity inversion) also indicates the presence of an inner basin and sill with "some non-magnetic sediment cover" (Tinto et al., 2015). This inner basin may hold a considerable volume of ice-proximal sediment deposited since the grounding line has been close to its present location in the fjord. Assuming, for example, 30 m of sediment fill across the basin (approximately 10 x 20 km in size after Tinto et al., 2015) adds 14 km$^3$ of glacigenic sediment to the total volume and increases the estimated average erosion rate to 0.39-0.45 mm a$^{-1}$. Additional material may also be stored subglacially upstream of the grounding line and observations from Greenland confirm that tens of meters of sediment is indeed present in places (Walter et al., 2014). However, previous studies of glacimarine sediment volumes from a range of Northern and Southern hemisphere fjords assume that the change in storage is negligible compared to the volume of material delivered to the fjord, particularly over 10$^2$-10$^3$ year timescales (Hallet et al., 1996; Koppes and Hallet, 2002; Fernandez et al., 2016) and we rely on the same assumption here. Nevertheless, for this reason, and because we cannot quantify the amount of sediment that exits the system in icebergs, our estimate should be taken as a *minimum* glacial erosion rate for the Petermann system.

## 5 Discussion

### 5.1 Sedimentary infill of the Petermann-Nares Strait system

During the past ~10 ka, the Petermann Fjord and adjacent parts of Nares Strait have been variably infilled with glacial marine sediment (Fig. 12) that is related primarily to the retreat of marine-terminating glaciers through the area, as is typical of today's ice-influenced fjords (Syvitski & Shaw, 1995). Stratigraphic, geomorphological and sedimentological observations are combined here with existing deglacial chronologies to produce a simple model for the evolution of fjord infill history. This model has three stages illustrating the major changes in glacier configuration, sediment supply and sedimentary processes.

1. During the LGM, Petermann Fjord, Hall Basin, and the Robeson and Kennedy Channels were all occupied by grounded ice of the coalesced Greenland and Innutian ice sheets (England et al., 2006 and references therein; Jakobsson et al., 2018). Subglacial lineations were produced by fast-ice flow in both the fjord and Kennedy and Robeson Channels depositing a thin till unit (e.g., Fig. 7c) that, based on our acoustic data, has a patchy distribution (e.g., Fig. 12a). However, till was also plastered onto bedrock terraces in the fjord as well as bedrock highs in Hall Basin (see Section 4.2).

2. By 8.7 cal. ka, when the grounding line of the Petermann Ice Stream was on the fjord-mouth sill and was probably fronted by an ice shelf (Jakobsson et al., 2018; Jennings et al., 2018), Hall Basin was in an ice-proximal setting (i.e., within several tens of kilometres from the grounding line). Similarly, outer Hall Basin and northern Kennedy Channel must have been in an ice-proximal setting when the ice margin in Nares Strait was located at the Kennedy Channel GZW, although the timing and duration of this stillstand event are not yet known. Subglacial deposits accumulated to form the wedges during the stillstands (Fig. 4). In Kennedy Channel, thick (>25 m) GFDs were deposited in front of the GZW (Figs. 10c, d, 12b) but similar units are conspicuously absent beyond the Petermann GZW (Figs. 8b, 12a). Seaward of these clear ice-proximal deposits, sedimentation in Hall Basin, Robeson Channel and northern Kennedy Channel, was largely from the settling of IRD and fine-grained material from meltwater plumes. On slopes greater than about 5° in Hall Basin, this material was redeposited down slopes and focussed into small basins (Fig. 12b), showing the strong influence of the rugged seafloor topography there. We find no evidence for additional stillstand events in our geophysical datasets, for example, on the S4 high between Hall Basin and Robeson Channel. As our understanding of ice shelf sediments increases (see Jennings et al., 2018; Smith et al., *Accepted*), clues to the proximity of the grounding line, the configuration of ice shelves, and the timing of stillstand/ice-shelf durations will be borne out by detailed (and forthcoming) sedimentological analyses of cores from Hall Basin.

3. Rapid glacier retreat through Petermann Fjord occurred after ~7.6 cal. ka (Jakobsson et al., 2018) with a floating tongue established over the fjord by 6.9 cal. ka (Reilly et al., 2019). This new ice configuration removed glacial sediment inputs from outer and mid-fjord areas, placing these areas in an ice-distal setting where deposition was mostly from settling of IRD and distal plume deposits. From this time, the tributary glaciers that enter the fjord through valleys incised into its sidewalls (see Fig. 2) became an additional source of glacial sediment to the fjord and have built up small ice-proximal fans in front of their marine margins (Fig. 12a; Fig. 6b of Jakobsson et al., 2018). The near instantaneous removal of sediment sources from the fjord (as the Petermann Ice Stream retreated rapidly) explains the lack of discrete accumulations of ice-contact or ice-proximal deposits in Petermann Fjord that are often associated with fjord settings (Fig. 12a; e.g., Hjelstuen et al., 2009; Stoker et al., 2009).

## 5.2 Comparisons with other ice stream systems: glacial sediment volumes and fluxes

There are relatively few previous studies that derive glacial sediment volumes, fluxes or basin-scale erosion rates for Greenland, and none (for erosion rates) that we are aware of that use volumetric analyses in fjords. While it remains difficult

to directly compare our results with other systems, recent mapping campaigns in the palaeo-catchment area of the NEGIS ice stream in North-East Greenland (Roberts et al., 2017) will allow for a similar detailed study of that system. One possibly

unusual feature of the Petermann Fjord-Nares Strait system is the absence of any thick (several hundreds of metres) accumulations of ice-proximal sediments beyond the fjord-mouth sill when the ice margin is known to have stabilised there for a period during retreat. As an analogue, a basin in front of the Jakobshavn Isfjord fjord-mouth sill holds more than 250 m of ice-proximal material deposited when the ice margin was at the sill (Hogan et al., 2012; Streuff et al., 2017) during the Fjord Stade c. 10.6-9.4 ka (Young et al., 2013; Streuff et al., 2017). Similarly, fjords in Norway, East Greenland and Patagonia are

known to contain 100-500 m of deglacial infill (Aarseth, 1997; Andrews et al., 1994; Bellwald et al., 2016; Fernandez et al., 2016) and seismic profiles of the inner shelf basin at the modern Pine Island Glacier ice shelf edge reveal that it holds >300 m of presumed ice-proximal sediment (Gohl, 2010; Nitsche et al., 2013). Given the similarity in fluxes between the palaeo Jakobshavn and Petermann ice streams, we suggest that lack of thick basin fill at Petermann is due to a shorter period of stabilization there; increased trapping efficiency of the large basin in front of the Jakobshavn sill when compared to the seafloor

morphology of Hall Basin; or some combination of both factors.

Our calculated sediment flux for the palaeo Petermann Ice Stream (1080-1420 $m^3$ $a^{-1}$ $m^{-1}$; uncertainty range 890-2170 $m^3$ $a^{-1}$ $m^{-1}$) is between estimates for modern ice streams (typically ~$10^2$ $m^3$ $a^{-1}$ $m^{-1}$; Kamb, 2001; Englehardt and Kamb, 1998; Anandakrishnan et al., 2007; Christoffersen et al., 2010) and those for the largest Norwegian palaeo-ice streams that delivered sediment to the shelf break (6000-11000 $m^3$ $a^{-1}$ $m^{-1}$; Nygård, 2003; Nygård et al., 2007). The calculated flux range is notably

similar to the range provided by Hogan et al. (2012) using the same methods for the palaeo-Jakobshavn Isbræ (1030-2300 $m^3$ $a^{-1}$ $m^{-1}$) when that ice stream was also stable at its fjord-mouth sill, although that estimate did not include a subglacial (coarse/mixed-grain size) component. During the LGM, these two ice streams operated with the same glacier thermal regime (i.e., warm-based streaming ice; Roberts and Long, 2005; Ó Cofaigh et al., 2013; England, 1999; Jakobsson et al., 2018), which is known to be a primary control on glacial erosion rates along with climate (Hallet et al., 1996; Koppes et al., 2015).

These two factors dominate over other variables like ice cover, sliding speeds and even ice flux (Elverhøi et al., 1998; Koppes et al., 2015), which explains the comparable estimates despite the larger (albeit modern) ice discharge of Jakobshavn Isbræ compared to Petermann (cf. Rignot and Kanagaratnam, 2006; Enderlin et al., 2014). The nature of the substrate is also important when considering sediment fluxes (Hallet et al., 1996) but its effect is somewhat difficult to assess for the two systems. Jakobshavn Isbræ erodes into banded gneiss with variable foliation and jointing (Roberts & Long, 2005) whereas

Petermann Fjord has been eroded into bedded limestones of lower Palaeozoic age (Dawes et al., 2000) with slabs being removed along bedding planes. The bedrock steps left by removal of limestone beds are visible in the seafloor morphology (Fig. 2; Jakobsson et al., 2018). Upstream of the bedded limestones, the bedrock is the typical Archaean crystalline basement of Greenland that includes gneisses and granitoids (Henriksen et al., 2009). The abrasion strength of these rock types (based on Schmidt hammer rebound values; Krabbendam and Glasser, 2011) are similar if the limestones are hard (Goudie, 2006),

but jointing is a major control on glacial plucking (e.g., Sugden et al., 1992; Dühnforth et al., 2010). Thus, it is difficult to distinguish between the erodability of bedded limestones and Archaean basement versus jointed gneiss. Numerical modelling

of these systems over our mapped bedrock surfaces and replicating our glacial fluxes would elucidate which factors control subglacial erosion rates and transport in the Petermann system. However, given the comparability of the two glacier systems, it appears from our results that the Petermann Ice Stream was approximately as efficient as the palaeo-Jakobshavn Isbræ at eroding, transporting and delivering sediment to its margin during the early deglaciation.

## 5.3 Comparisons with other fjord systems: glacial erosion rates

Erosion rates (and sediment fluxes) are likely to vary during a glacial-deglacial cycle due to pulsed ice streaming (e.g., Christoffersen et al., 2010) and because, early in the deglacial period, ice streaming may have been over unconsolidated sediment recently deposited during the preceding glacial advance (Elverhøi et al., 1998). Furthermore, increased erosion rates have been correlated with higher ice velocities associated with recent glacial retreat (Koppes and Hallet, 2002, 2006; Koppes et al., 2009). Our glacial erosion values for an earlier and later phase of deglaciation (1.41-1.85 mm a$^{-1}$ and 0.14 mm a$^{-1}$, respectively) indicate that deglacial erosion rates may have been an order of magnitude larger during the early deglacial when Petermann Ice Stream was grounded on the sill. Presumably, during this earlier stage, thinning ice and warmer basal temperatures led to enhanced ice flow at the bed (cf. Koppes and Montgomery, 2009); likewise, the ice stream was also in an expanded state allowing for relatively high erosion rates. Furthermore, there is landform evidence that surface meltwater may have reached the bed at this time (Jakobsson et al., 2018) thereby increasing the potential for subglacial erosion. We have to acknowledge that some sediment in inner Hall Basin may have been produced by ice in Kennedy Channel, rather than the Petermann Ice Stream, which would artificially raise the early phase erosion rate calculated here. It is not possible to separate these two components based on currently available information, meaning the early phase erosion rate may be overestimated. However, our results are in line with past work showing that glacial erosion rates vary significantly over different timescales (cf. Koppes and Montgomery, 2009) and with different glaciologic states, especially during retreat when the glacier system experiences rapid changes (e.g., Hallet et al., 1996; Koppes and Hallet, 2002, 2006).

There are relatively few estimates of glacial erosion rates from Greenland. For the Kangerdlugssuaq Fjord and Trough system in East Greenland, Cowton et al. (2012) updated the modern erosion rate of Andrews et al. (1994) from 0.01 mm a$^{-1}$ to 0.3 mm a$^{-1}$. The former was based on estimated sediment discharges (for a certain ice flux) and Cowton et al. (2012) included the sediment deposited beneath the mélange (after Syvitski et al., 1996). However, as Cowton et al. noted, the Andrews et al. (1994) study assumed that glacial erosion occurred over the entire Kangerdlugssuaq catchment area (~50 000 km$^2$) including a large part of the ice-sheet interior which has very low velocities (cf. Rignot and Kanagaratnam, 2006; Howat, 2017). In our comparison of glacial erosion rates, we elect to exclude portions of the ice sheet interior that are likely frozen to the bed and, accordingly, decrease the catchment area for Kangerdlugssuaq to areas with ice velocities that would permit subglacial erosion. (Note that the catchment areas used for erosion rate calculations are fully described in Supplementary Information and Fig. S1). Using a catchment area (9437 km$^2$) that includes only ice flowing at >50 m a$^{-1}$ for the Kangerdlugssuaq system (Fig. S1c), as we have applied at Petermann, the modern erosion rate for the Kangerdlugssuaq system becomes 1.46 mm a$^{-1}$. This rate is about three times larger than the average deglacial rate for the Petermann system. A useful exercise may be to calculate the

basin-wide deglacial erosion rate for the Jakobshavn catchment area using the volume of glacimarine sediments deposited in front of the fjord-mouth sill (29.2 km$^3$) during an 800 year stillstand (Hogan et al., 2012) and a glacial catchment area derived using the same procedures in this study (33 504 km$^2$; Fig. S1b). This returns a glacial erosion rate for the palaeo-Jakobshavn Isbræ of 0.52 mm a$^{-1}$ that can be compared with the early deglacial erosion rate for Petermann (1.41-1.85 mm a$^{-1}$), as this was also calculated for the time when the grounding-line was stable at its fjord mouth. As both systems were drained by a single,

large, fast-flowing ice stream during the last glacial, the lower values for the palaeo-Jakobshavn ice stream may simply reflect the larger drainage basin used in those calculations. We note that the area of fastest-ice flow (>400 m a$^{-1}$) is considerably larger in the Petermann system than the Jakobshavn system (Petermann Fjord is about twice as wide) and that rates of glacial erosion are up to four times higher in fjords compared with inter-fjord areas (Stroeven et al., 2002; Briner et al., 2006). If the majority of glacial erosion occurs only in these narrow corridors for major outlet glacier systems, then the calculated glacial erosion

rates would differ significantly as the narrow geometry of Jakobshavn would produce a much higher erosion rate. This indicates the need for a careful and consistent approach to defining the effective drainage basin area in glacial erosion studies for major outlet glaciers.

      Modern glacial erosion rates have also been provided for the well-studied Kangerlussuaq area in central West Greenland, by measuring annual sediment loads (suspended and in solution) in proglacial rivers beyond land-terminating glaciers (Cowton

et al., 2012; Hawkings et al., 2015; Hasholt et al., 2018) and dividing by the catchment area. Although individual study years have returned rates as high as 4.5 mm a$^{-1}$, for the decade 2006-2016 the average rate was 0.5 mm a$^{-1}$ (Hasholt et al., 2018). These studies used a consistent approach to defining the catchment area based on the ablation area for the Kangerlussuaq drainage basin and modelled hydrological catchment, which we deem as comparable to the approach taken here (i.e., excluding portions of the ice-sheet interior where erosion is limited). The average modern erosion rate from Kangerlussuaq (0.5 mm a$^{-1}$)

is similar to our average deglacial erosion rate for Petermann (0.29-0.34 mm a$^{-1}$) despite the differences in methodologies employed, timescales studied (millennial vs. annual/decadal) and the glaciologic setting (multiple land-terminating glaciers vs. one large marine-terminating ice stream). Regarding the latter, significant surface melt occurs at Kangerlussuaq that then migrates to the bed via moulins and entrains sediment as it drains subglacially (Cowton et al., 2012). In contrast, although supra-glacial lakes are documented on the grounded portion of the modern Petermann Glacier during the summer, and may

drain to the bed (MacDonald et al., 2018), the fast flow is the dominant control on basal sliding (cf. Nick et al., 2012) and, therefore, presumably on glacial erosion for this catchment. Previous studies have suggested that modern rates may not be representative of longer-term (millennial) rates because of recent increases in subglacial erosion (and/or sediment evacuation) as glaciers accelerate in today's warming climate (Koppes and Montgomery, 2009). This is certainly true for the GrIS where surface mass balance has become increasingly negative over the last four decades (Mouginot et al., 2019) suggesting that

modern glacial erosion rates have probably started to rapidly accelerate over the last decade. However, the rates that we calculate for the Petermann system are for a major phase of deglaciation when the ice stream likely accelerated and subglacial erosion was enhanced, and therefore may be comparable to accelerated retreat of today's glaciers. Regardless, we must be cautious when comparing rates that employ different procedures and are determined for very different timescales.

There is a large body of previous work using the volume of glacimarine sediments in fjords to derive sediment yields and, ultimately, glacial erosion rates during retreat (e.g., Powell, 1991; Hunter, 1994; Stravers and Syvitski, 1991; Hallet et al., 1996; Elverhøi et al., 1995; Koppes and Hallet, 2002; Fernandez et al., 2016). Erosion rates for Alaskan glaciers, where the climate is temperate and tectonic uplift are major contributing factors, are exceptionally high (>10-100 mm a$^{-1}$; Hallet et al., 1996). The study of Fernandez et al. (2016) reported average millennial erosion rates between 0.02 and 0.83 mm a$^{-1}$ for Patagonian and Antarctic Peninsula fjord systems (since deglaciation) and provides a ready comparison to the results of this study. Their values for the Antarctic Peninsula cluster around 0.1 mm a$^{-1}$, which is comparable to the average value we derive for the Petermann catchment. They also highlight a decrease in erosion rates with increasing latitude that they attribute to decreasing temperatures and availability of liquid water at the ice-rock interface. The Petermann area, situated at ~81° N, has a polar climate with a mean annual temperature (MAT) of around -11°C (for Thule airbase; yr.no, 2019) at present; based on reconstruction from ice cores, surface air temperatures were around 1-3°C higher than today during deglaciation (Lecavalier et al., 2017). The only system with a comparable MAT in the Fernandez et al. study is Herbert Sound on the Eastern Antarctic Peninsula (MAT = -7.8°C; $\bar{E}$ = 0.12 mm a$^{-1}$); however, as noted earlier, relatively little surface meltwater accesses the bed in this type of glaciologic setting and the fast-flow of feeder glaciers likely dominates glacial erosion. We suggest that the higher deglacial erosion rate at Petermann compared with the Antarctic Peninsula fjords was, therefore, most likely caused by a high trapping efficiency of the Petermann Fjord-Hall Basin setting in conjunction with the erosive potential of a major (~20 km wide; > 1500 m thick) ice stream draining the area during deglaciation.

## 6 Conclusions

We present the first comprehensive, high-resolution investigation of the glacial-sedimentary infill of a major fjord system in Greenland. The seismic stratigraphy of Petermann Fjord and the adjacent Nares Strait area confirm the episodic retreat of ice streams in the area marked by GZW deposits, followed by the deposition of sediment from meltwater plumes and icebergs. The rugged bedrock topography is a major control on sediment distribution in relation to the retreating ice margin; redeposition by gravity flows was important only on local scales. Our mapped unconsolidated sediment volumes provide glacial sediment fluxes for the former Petermann Ice Stream when it was stable on a sill at the fjord mouth that are in line with sediment flux estimates from modern Antarctic and other Northern Hemisphere palaeo-ice streams, including the palaeo-Jakobshavn Isbræ. The average deglacial erosion rate that we calculate for the Petermann drainage basin is one of only a few erosion rate estimates for Greenland; it is similar to the rates from the Antarctic Peninsula and some Patagonian catchments, despite being subject to a much colder climate. In this setting, ice dynamics such as the fast-flow of Petermann Glacier (or former ice stream), rather than climate, are the dominant controls on glacial erosion. The order-of-magnitude difference between glacial erosion rates during an early phase of deglaciation (when the grounding line was stable at the fjord mouth) and a later phase (of retreat through the fjord) confirm significant variability in erosion rates related to deglacial retreat rates and ice dynamics. Mapped pre-LGM surfaces, calculated glacial sediment fluxes, and our range of glacial erosion rates provide much needed

observational constraints on future numerical modelling experiments of the Petermann system, now one of the best studied outlet glacier systems in Greenland.

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

**Data availability**

The marine geophysical data used in this paper can be obtained by contacting the second author apart from the 2001 BGR seismic-reflection profiles which are available on request from: https://cdi.seadatanet.org/.

**Author contributions**

K.A.H., M.J. and L.M. conceived the study; they and B.R., A.J., A.M., K.H., E.N., K.J., and C.S. collected the data during the *Petermann 2015 Expedition*. T.N., K.J.A. and E.K. performed some initial mapping. K.A.H. analysed the SBP data, integrated

it with the seismic-reflection data and calculated flux and erosion estimates with contributions from M.J., A.J. and B.R. K.A.H. wrote the initial manuscript with substantial contributions from M.J., B.R., A.J. and L.M. All authors contributed to data interpretation and writing of the final manuscript.

**Competing interests.**

The authors declare no competing interests.

## Acknowledgements

The *Petermann 2015 Expedition* was supported by the US-NSF Polar Programs (Awards 1418053 to Mix and Stoner, 1417787 to Mayer, and 1417784 to Jennings), NASA, UNAVCO, CH2M Polar Field Services (Jessy Jenkins), Polar Geospatial Center, the Swedish Polar Research Secretariat and the Swedish Maritime Administration. We thank the Federal Institute for Geosciences and Natural Resources (BGR), Hannover for permission to use the 2001 seismic-reflection profiles in this study. KAH's time was supported by the British Antarctic Survey "Polar Science for Planet Earth" programme funded by the UK's Natural Environment Research Council, and by the Visiting Scholar Program at the Center for Coastal and Ocean Mapping at the University of New Hampshire. MJ and colleagues from Stockholm University were supported by a grant from the Swedish Research Council (VR). We also thank *Geocenter Danmark* who supported the inclusion of multichannel seismic data acquisition during the *Petermann 2015 Expedition* and Per Trinhammer for his help acquiring these data. We thank Neil Arnold for useful discussions about Greenland glaciology/hydrology.

| Core name | Latitude (°N) | Longitude (°W) | Water depth (m) | Length (cm) | Cruise / reference |
|---|---|---|---|---|---|
| OD1507-37PC | 80.96575 | 60.95450 | 1041 | 847.6 | *Petermann 2015 Expedition* |
| OD1507-41GC | 81.19378 | 61.97715 | 991 | 440 | *Petermann 2015 Expedition* |
| OD1507-52PC | 81.24183 | 63.99833 | 517 | 541 | *Petermann 2015 Expedition* |
| HLY0301-05GC | 81.62143 | 63.25778 | 797 | 371 | HLY0301 / Jennings et al. (2011) |

**Table 1. Core locations and acquisition information for cores used in seismo-acoustic facies and lithofacies correlations.**

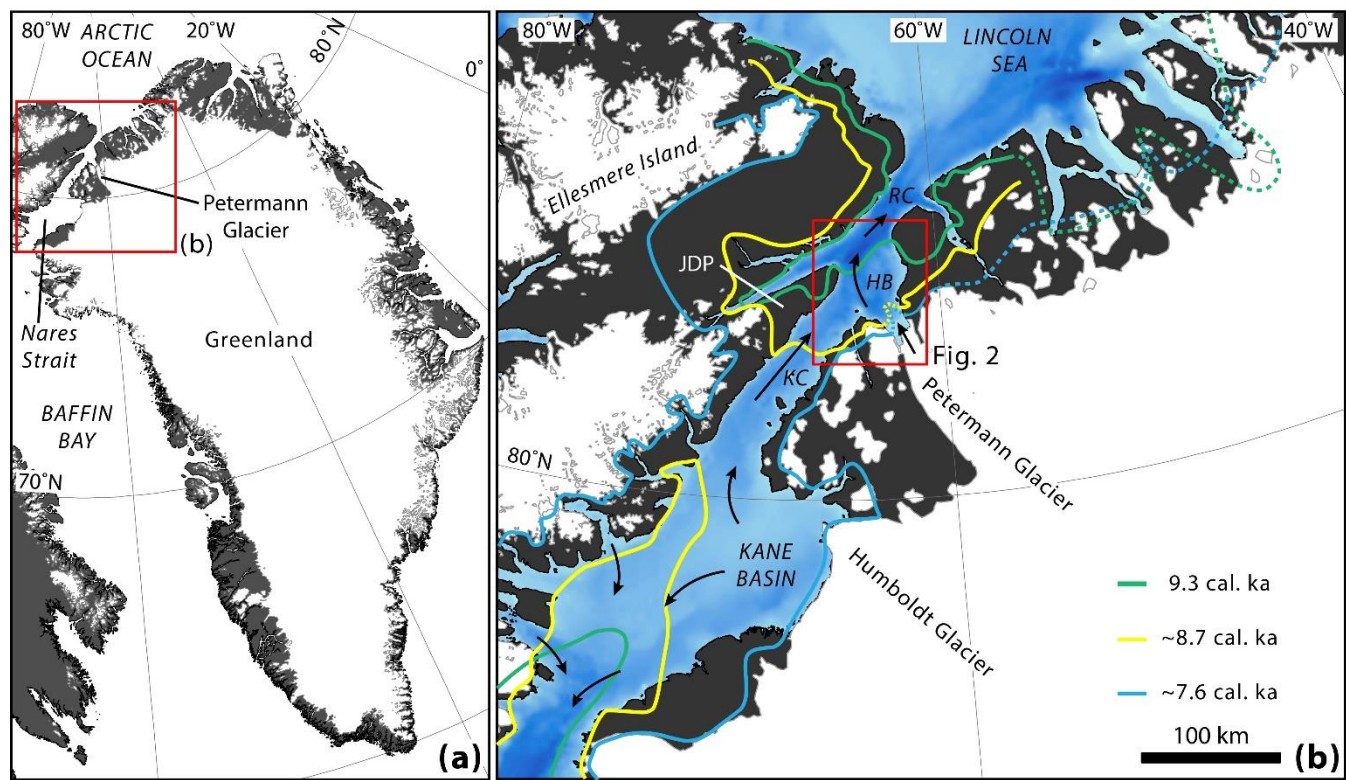

Figure 1. Maps of Greenland and the study area. (a) Location of Petermann Glacier and Nares Strait. (b) Location of the study area (red box; Fig. 2) in Petermann Fjord and the adjacent Nares Strait including Hall Basin (HB), Kennedy Channel (KC) and Robeson Channel (RC); JDP is Judge Daly Promontory. Ice flow in marine areas (black arrows) and deglacial ice-sheet margins for the early Holocene (9.3 cal. ka, ~8.7 cal. ka, ~7.6 cal. ka) are also shown for Nares Strait and were compiled from England (1999), Georgiadis et al. (2018), Jakobsson et al. (2018). Dashed lines outside of this area are from Young & Briner (2015) and references therein.



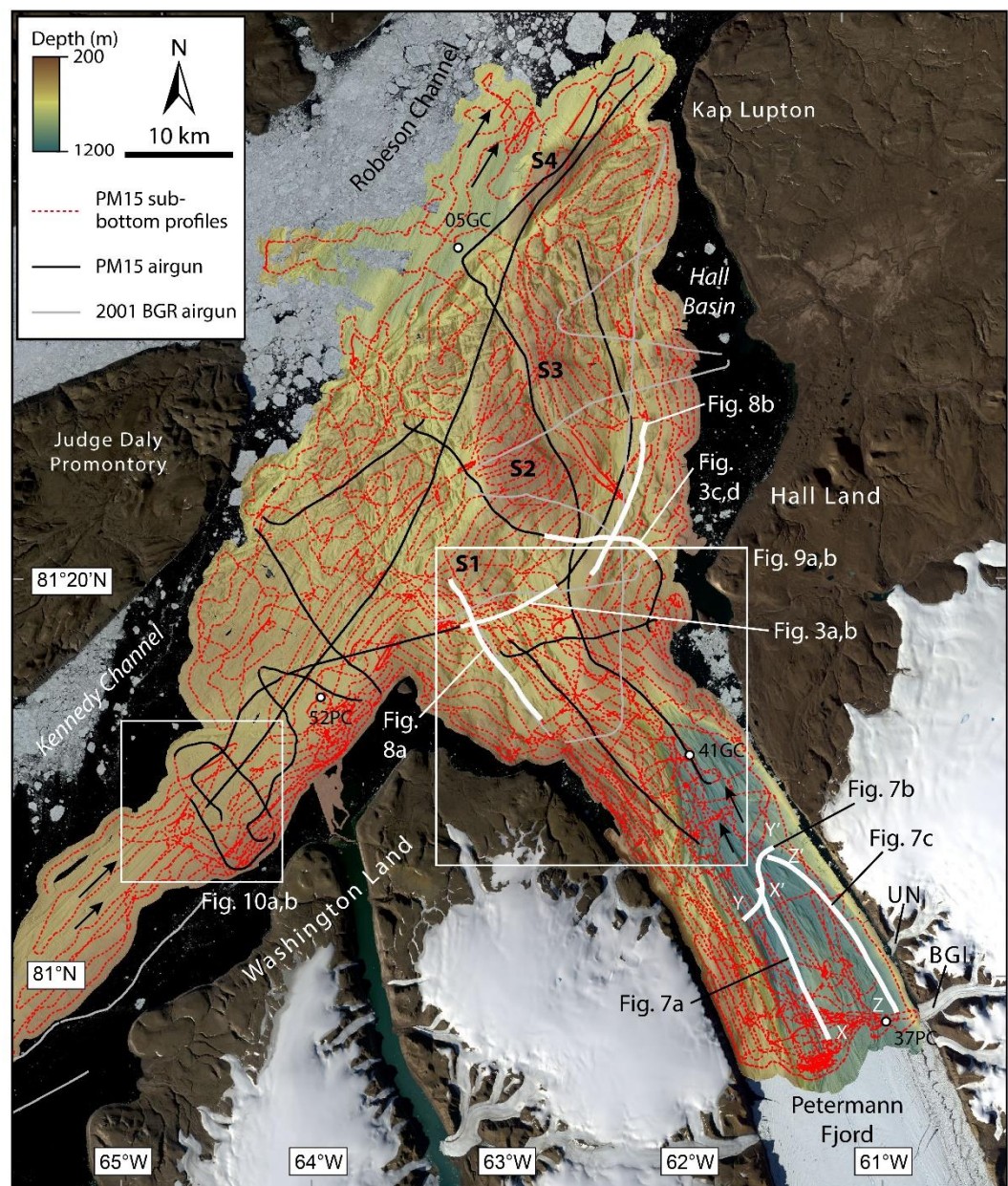

**Figure 2. The locations of SBP profiles (red dashed), 2015 AG profiles (black), 2001 legacy AG profiles (grey) shown over the gridded multibeam bathymetry for the area. S1-S4 are the bathymetric highs described by Jakobsson et al. (2018) and referred to in the text. UN is Unnamed Glacier; BGl is Belgrade Glacier. Glacial lineations denoting the former directions of ice flow are shown as black arrows (after Jakobsson et al., 2018). Sediment cores used to correlate seismo-acoustic facies with sediment lithofacies are also shown.**


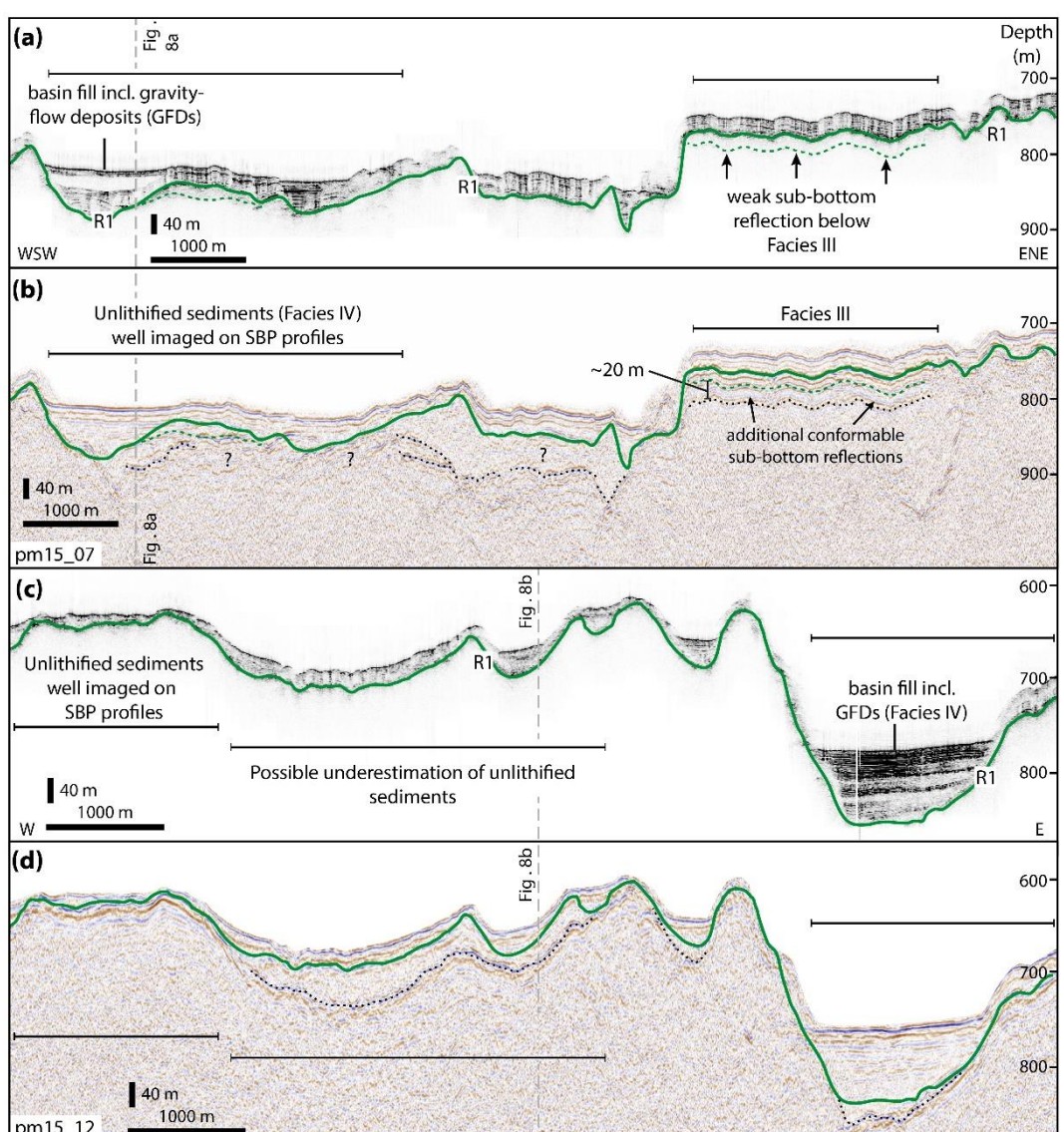

**Figure 3. Comparison of SBP profiles with coincident AG profiles showing the mapped basal reflector (R1) on SBP profiles and the corresponding reflector (green) on AG profiles. Dotted lines mark the lowermost reflections on each profile; dotted lines and question marks indicate uncertainty in mapping of unlithified sediment package over basement. Sub-bottom reflections may be geological boundaries in sedimentary bedrock in Hall Basin. (a) SBP profile acquired on 11th August, 2015 in Hall Basin, coincident with AG profile pm15_07 (b). (c) SBP profile from 18th August, 2015 also in Hall Basin, coincident with AG profile pm15_12 (d). Location of profiles is shown in Figure 2; intersection of Fig. 6b with (c) and (d) shown as vertical grey dashed line.**



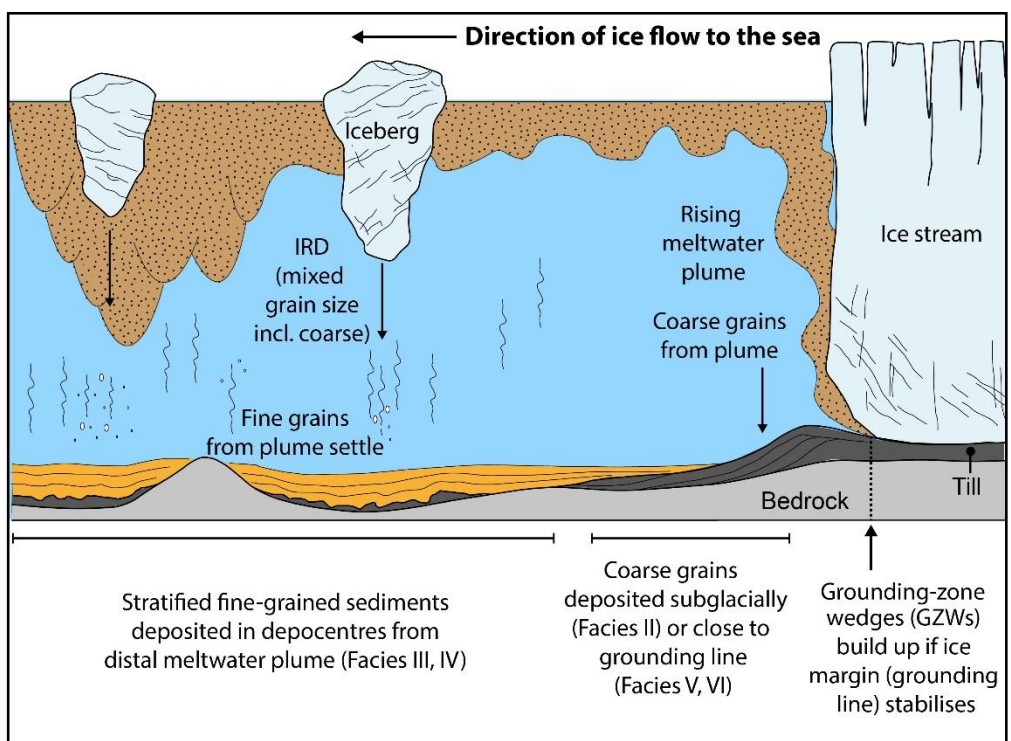

**Figure 4. Processes of glacimarine sedimentation at the marine-terminating margin of a Greenland outlet glacier (no ice shelf/tongue). The related seismo-acoustic facies as mapped in the Petermann-Nares Strait system are shown at the bottom of the figure.**


| Acoustic facies and configuration | Thickness (m) | Sub-bottom profiler or airgun profile |
|---|---|---|
| *I. Acoustically-impenetrable to homogenous* Moderate to high-amplitude prolonged, continuous reflection defining a rugged surface geometry. Rare sub-bottom point / hyperbolae reflections. | N/A | 20 m |
| *II. Acoustically-homogenous to transparent, non-conformable* Prolonged upper reflection and discontinuous weak basal reflection. Variable thickness and unconfomable with underlying reflections. Homogenous or transparent. | 5 - 20 | 20 m |
| *III. Acoustically-stratified conformable* Medium amplitude, parallel reflections, high continuity, conformable geometry. High-amplitude, continuous upper and basal reflections. | 5 - 15 | 5 m |
| *IV. Acoustically-stratified basin fill* Medium amplitude, parallel reflections, high continuity, basin fill (ponded) or onlapping geometry. Occasional thicker transparent units. Continuous upper and basal reflections. | 5 - 35 | 10 m |
| *V. Acoustically-transparent / semi-transparent* Low to medium amplitude reflections surrounding acoustically transparent to semi-transparent bodies in basins or on slopes. Geometry is lenticular to variable thickness tapering at one or both ends. | 2 - 25 (lenses 2 - 10 m thick) | 10 m |
| *VI. Downlapping to chaotic* Low amplitude, discontinous reflections with a disorganised downlapping or chaotic geometry. Only observed on AG profiles. | <150 | 20 m |

**Figure 5. Seismo-acoustic facies identified from SBP profiles and AG profiles in Petermann Fjord and Nares Strait. Seismo-acoustic facies I-V mapped primarily on SBP profiles and checked with AG profiles; facies VI mapped only from AG profiles.**


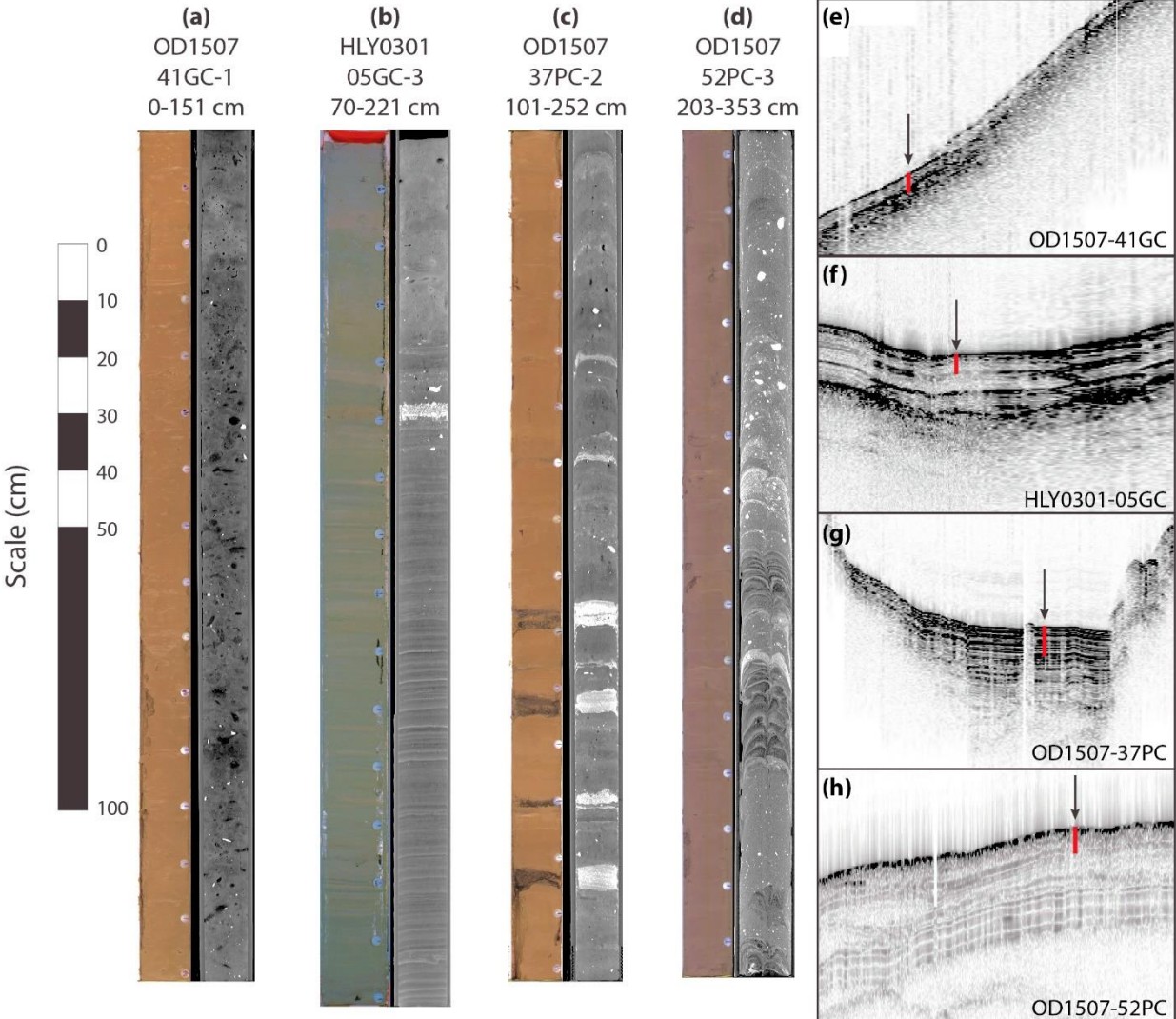

**Figure 6. Correlation of core lithofacies with seismo-acoustic facies III (a, b, e, f), IV (c, g) and V (d, h). Core lithofacies are shown using CT-scans (dark = lower density; light = higher density) and core photographs; seismo-acoustic facies at each core site are shown in panels e-h. Note coring artefact in core OD1507-52PC with layers bent down at the sides. The red bars in panels e-h represent the total core length for each core (using a velocity through sediment of 1500 m s⁻¹); core locations are shown on Figure 2.**


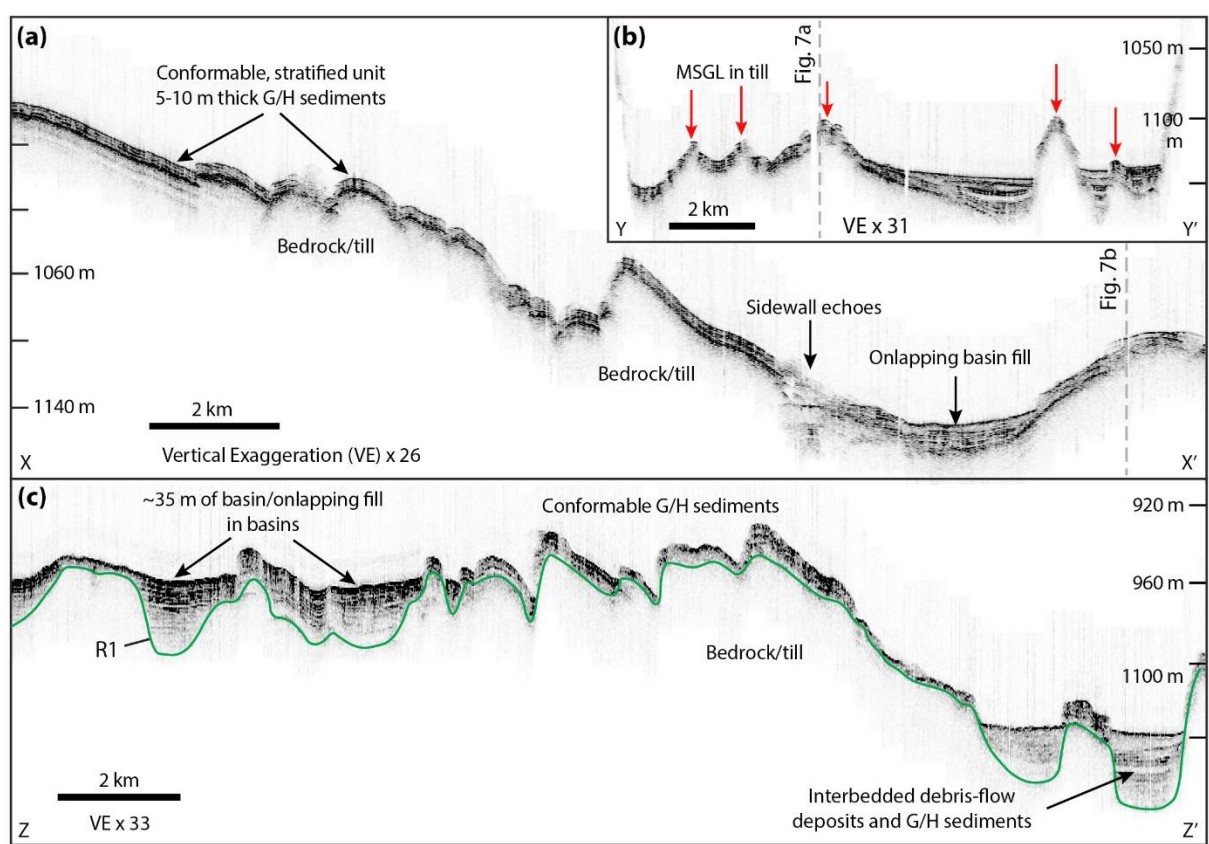

**Figure 7. Typical SBP profiles from Petermann Fjord (see Fig. 2 for locations) showing the acoustic stratigraphy of the glacimarine sediment package. (a) Fjord-parallel line showing conformable units (*Facies III*) overlying R1 reflection. (b) Outer fjord profile running approximately SW-NE showing conformable fill (*Facies III*) over subglacial till deposits (*Facies II*) mapped as MSGL (red arrows) and basin fill with GFDs (*Facies IV*) in local depressions. (c) Fjord-parallel line on the eastern side of the fjord showing basin fill in local depressions and conformable fill elsewhere. G/H is glacimarine/hemipelagic sediments.**


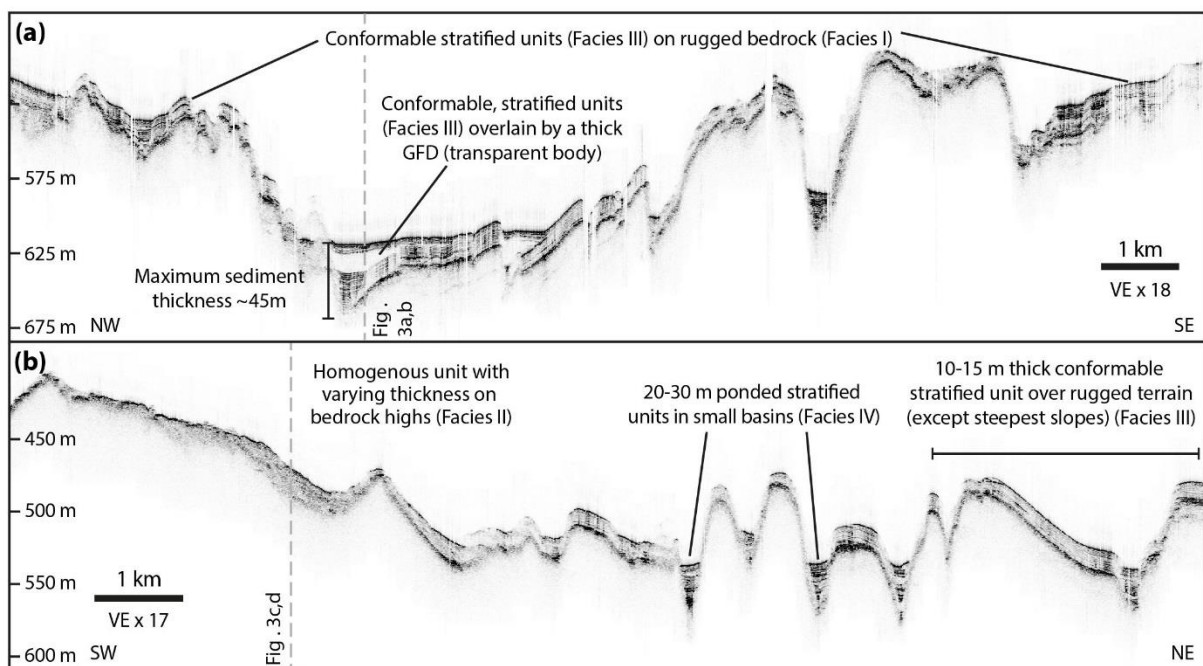

Figure 8. Examples of SBP profiles from Hall Basin, Nares Strait (see Fig. 2 for locations). (a) NW-SE profile in Hall Basin showing bedrock topography (*Facies I*) mantled with conformable sediment (*Facies III*) and ponded basin fill, sometimes with significant GFDs in local depressions (*Facies IV*). (b) A SW-NE profile between the Petermann sill and S1 high showing a similar stratigraphy but including non-conformable, homogenous sediment on steep slopes (*Facies II*). Intersections with Figs. 3a-d are marked with vertical grey dashed lines.

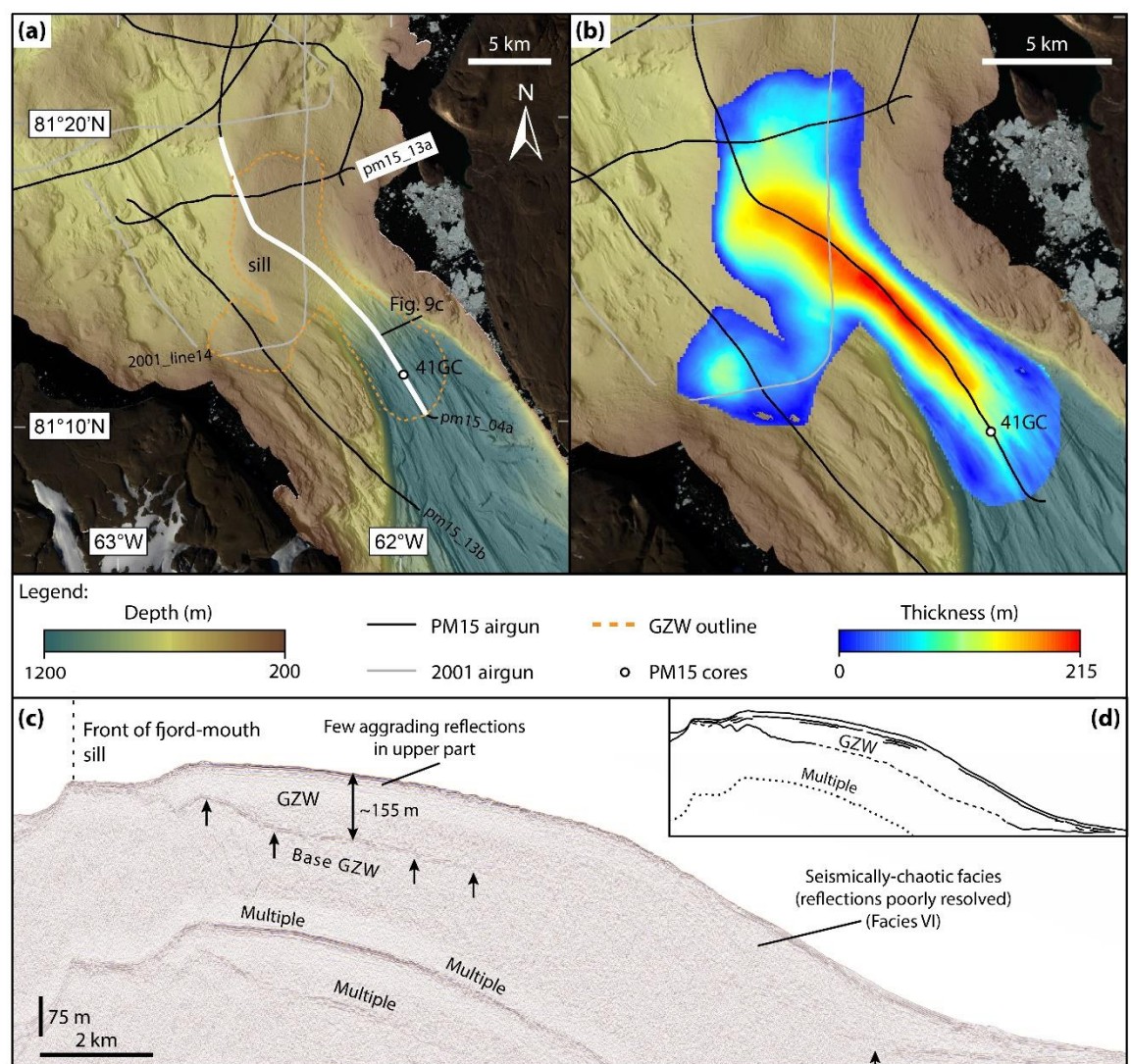

**Figure 9. Mapping of the Petermann sill grounding-zone wedge (GZW). (a) AG profiles over the GZW and outline used in volume calculations. (b) Isopach map of the GZW based on mapping from AG lines. (c) AG profile pm15_04a showing the seismic stratigraphy (*Facies VI*) of the GZW. (d) Line drawing of AG profile pm15_04a.**


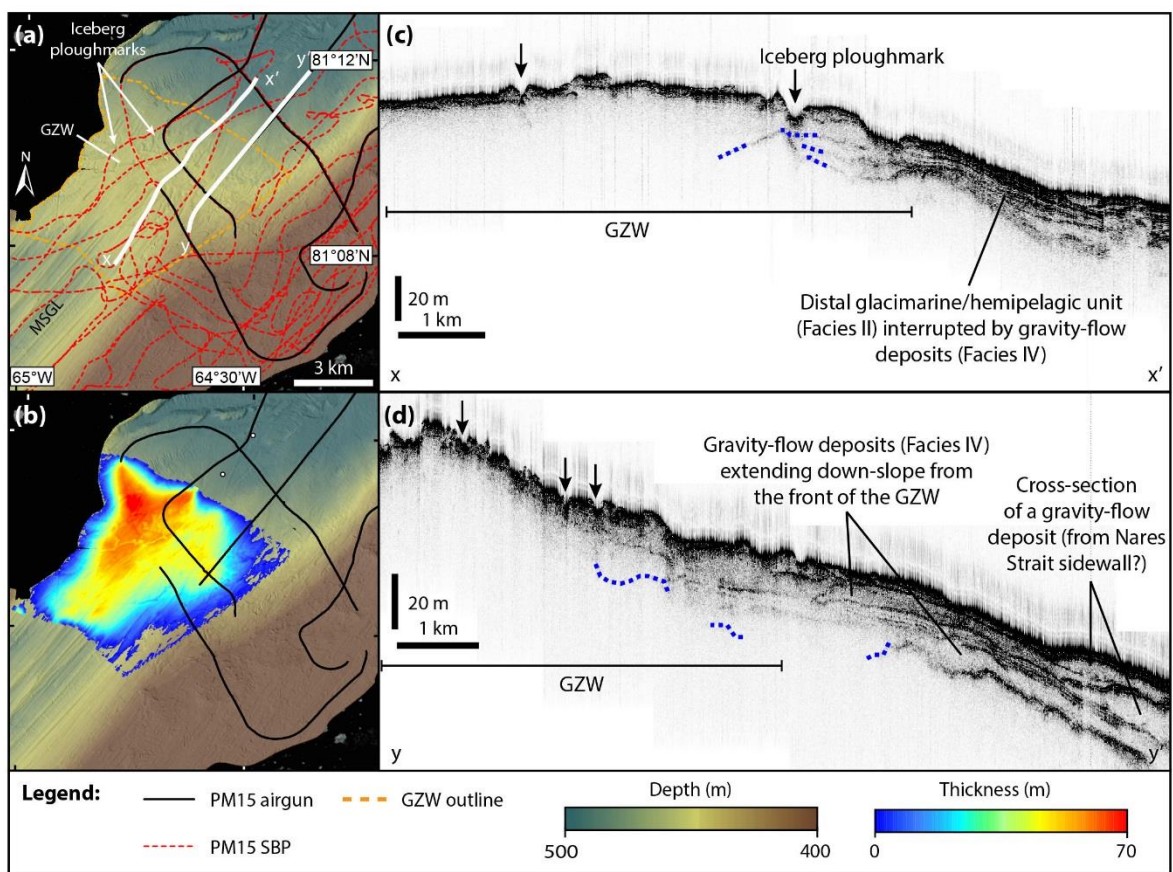

**Figure 10. Mapping of the Kennedy Channel GZW. (a)** AG and SBP profiles over the GZW and outline used for volume calculations. **(b)** Isopach map for the Kennedy Channel GZW (using a sound velocity of 1500 m s⁻¹). **(c)** SBP profile over the GZW showing the acoustically semi-transparent lenticular bodies (*Facies IV*) interfingered with acoustically stratified conformable units down slope (*Facies III*); location shown in (a). **(d)** SBP profile of the frontal part of the GZW showing semi-transparent units tapering down slope (*Facies IV*); location shown in (a). Black arrows point to iceberg ploughmarks; blue dashed lines show deepest sub-bottom reflections in the GZW interpreted as the base of the GZW.


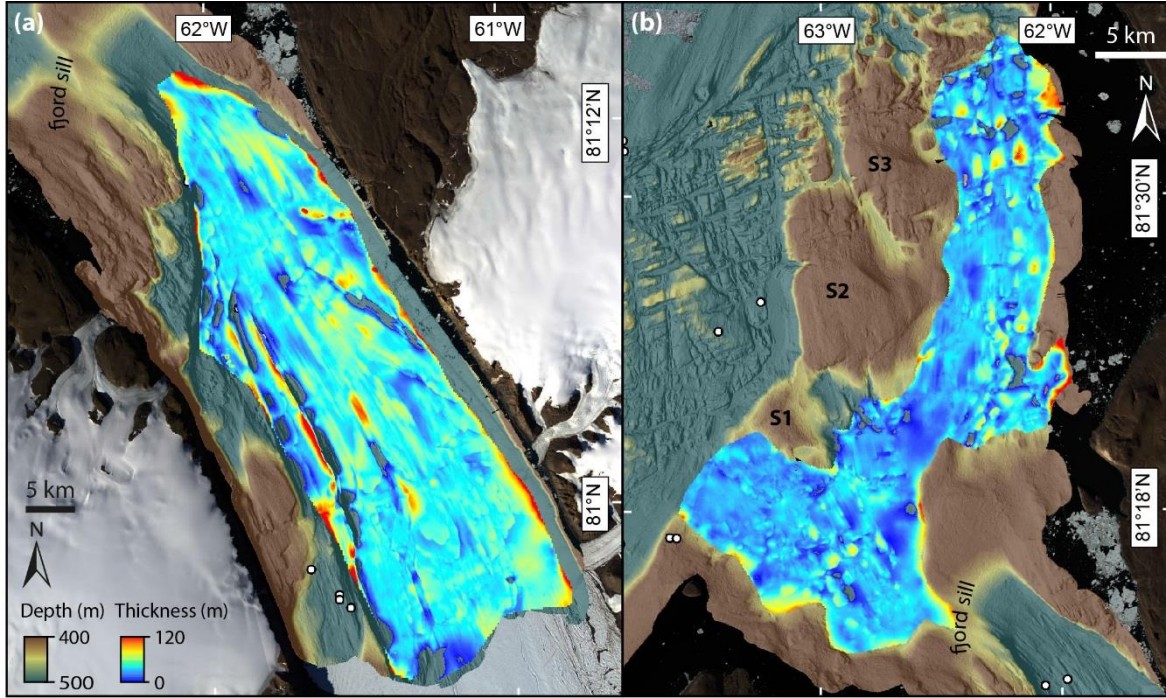


**Figure 11. Isopach maps of the deglacial sediment pile for (a) Petermann Fjord, and (b) inner Hall Basin.**

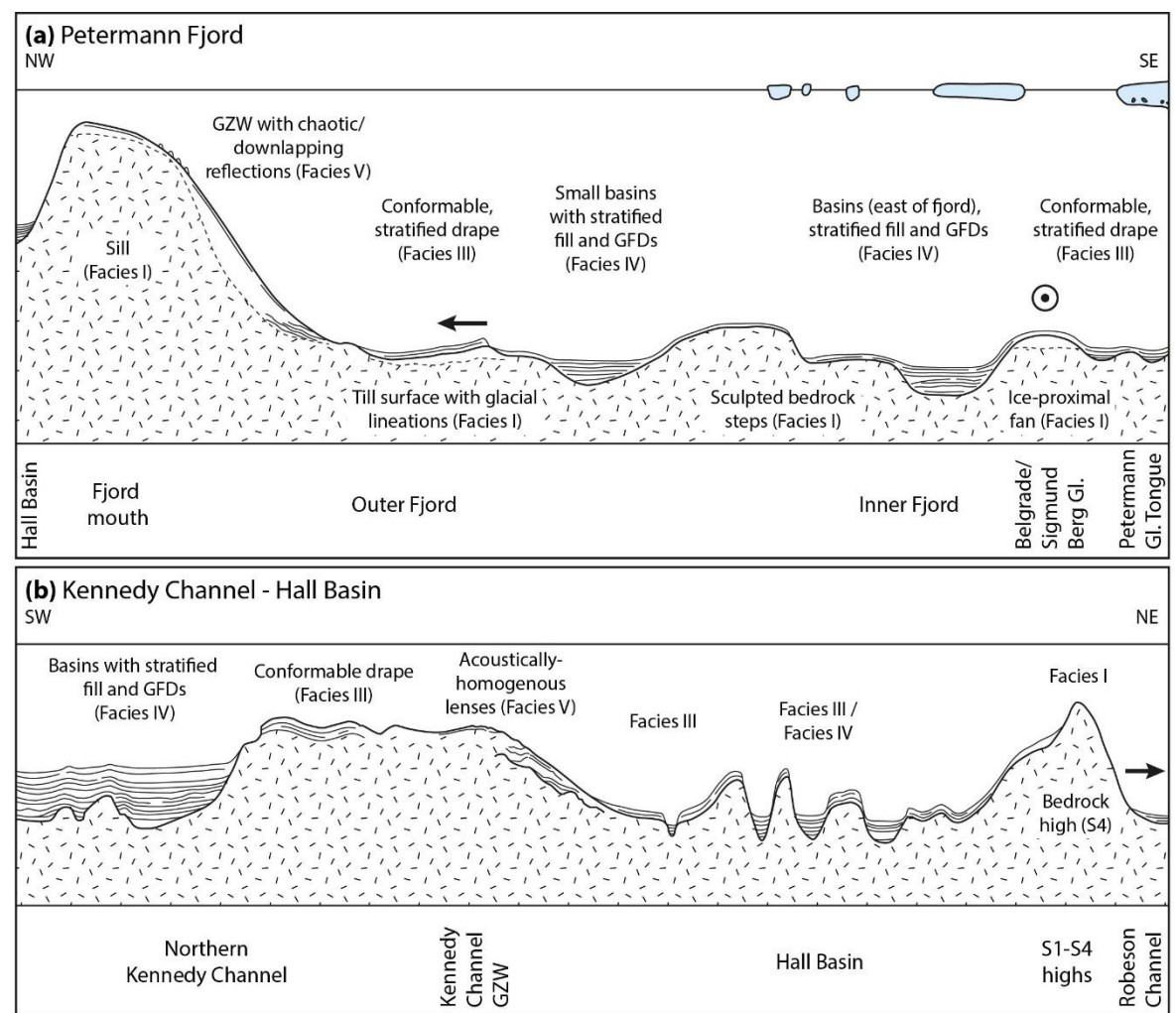

**Figure 12. Conceptual transects showing the seismic stratigraphy and distribution of glacimarine sediments in the Petermann Fjord-Nares Strait area. (a) Petermann Fjord with the fjord-mouth sill with GZW on the left side; localized sediment input into the NE side of the fjord from tributary glaciers and building an ice-proximal fan is shown as the bullseye. (b) Deglacial sediment cover in Nares Strait from Kennedy Channel to Hall Basin to Robeson Channel. Not to scale. Black arrows show the former ice flow direction through the system.**
