# Peer review of "Glacial sedimentation, fluxes and erosion rates associated with ice retreat in Petermann Fjord and Nares Strait, NW Greenland"

_The Cryosphere, 2019_

## Referee Comment (RC1) · Anonymous Referee #1 · 4 Sep 2019

This is a well written and illustrated paper that presents new data on glacigenic sediment fluxes and erosion rates associated with the retreat of Petermann Glacier in NW Greenland. The paper is very appropriate to The Cryosphere. However, there are a few issues that need to be addressed:

1. You state that this is the first comprehensive investigation of the glacial-sedimentary infill of a major fjord system in Greenland (last para of Introduction and first line of Conclusions). I think you could do with justifying this claim a bit more robustly. For example the lead author has herself published on a similar topic from Disko Bugt/Jakobshavns Isbrae (e.g., Hogan et al., 2012 Marine Geology) and there has been previous work

carried out on seismic stratigraphy in East Greenland fjords (Scoresby Sund – e.g., Uenzelmann-Neben, et al. 1991: Quaternary sediments in Scoresby Sund. East Greenland: their distribution from rellection seismic data. In Moller et al. (eds.): The Last Interglacial-Glacial Cycle: Jameson Land and Scoresby Sund, East Greenland). It may be that the present study significantly supersedes this earlier work but more justification/comment on this is required.

2. Line 132. Explain why a delta-R of 268+/- 82 was chosen.

3. I suggest you change Sub-heading 4.2 'Petermann Fjord' to something a bit more informative, particularly as sub-heading 4.1 is 'Seismo-acoustic facies and depositional environments in Petermann Fjord and Nares Strait'.

4. Lines 230-233 and Line 316. You state that subglacial tills are deposited as sediment gravity flows. But gravity flows are NOT subglacial tills. For a recent treatment see D.J.A. Evans 2018 – 'Till'. Please change.

5. Lines 369-371. You suggest that pre-LGM sediments most likely do not occur citing as support the presence of glacially-sculpted bedrock surfaces and referring to Jakobsson et al. (2018). I think this is pretty thin evidence. For example, pre-LGM sediments could be preserved locally within bedrock depressions and/or be too thin to be resolved by your seismic system(s). At the very least I would suggest the inclusion of a caveat acknowledging this would be appropriate here.

6. Lines 372-373. You state that glaciomarine sedimentation seaward of the grounding line has two components the first of which is that of coarse or mixed material delivered to the grounding zone as subglacial deposits. But how is this material actually deposited? It is glaciomarine sediment deposited seaward of the grounding line, so although it might be delivered to the grounding line subglacially it is not deposited seaward of the grounding line as a subglacial till.

7. Line 430. You mention previous work which suggests that most of the sediment

delivered by tidewater glaciers remains in the fjord system and you go on to state that "this assumption is particularly reasonable for Petermann". But is Petermann really a tidewater glacier? Surely it is more of a hybrid between an ice shelf and a tidewater glacier given its floating tongue?
* * *

---

## Referee Comment (RC2) · Anonymous Referee #2 · 11 Sep 2019

The manuscript of Hogan et al. present new knowledge from the Petermann Fjord, and is based on seismic profiles. The study gives important information from a fjord environment that rarely are surveyed due to ice conditions and its remote location. The paper is suitable for publishing in the journal, but only after a moderate to major revision. The main comment to the paper is that it must be better structured and that the discussion has also to implement a depositional history of the fjord system.

Comments:

1) Chapter 1. Introduction: The paper should better explain why this study is of regional interest. As it is in the present paper the focus is quite local (Greenland). Why should

we read this paper?

2) Chapter 1. Introduction: In the listing of the objectives (lines 70-75) a more overall objective should also be included, e.g. compare erosion rates in this fjord environment with fjords elsewhere, global outlook, regional considerations. The objectives as they are now is quite "local".

3) Some more references to Figures 1 and 2 should be included in Chapter 1

4) In the heading of Chapter 2 delete "(geology, physiography, oceanography)"

5) Chapter 3 should be renamed to "Data and Methods" as you also describe the data used in the study

6) Chapter 3 does not include information about the cores you have used in the study. Information about these cores (and which now partly are in Supplementary Material) should be mentioned.

7) All information provided in Supplementary Material should instead be implemented in the main text as I think the information in Supplementary Material is essential for the paper.

8) The methods used in the papers (how to calculate drainage area, erosion rates etc) are as I read it spread a bit around in the paper. Please structure the paper in such a way that all your methods are included in Chapter 3

9) The Result chapter should be restructured. As it is now it is some repetition of the text. I suggest that you in 4.1 included information (text and figures) from the Supplementary Material in defining your facies. Furthermore, I suggest that Chapters 4.2 and 4.3 are merged to get a better overall view of the study area. You do not need to repeat the definition of the facies in chapters 4.2 and 4.3 (or what these facies are composed of) since you have already defined this in Chapter 4.1.

10) Why do the manuscript have an own chapter on GZWs (Chapter 4.4). This chapter

could also be merged into the merged 4.2-4.3 chapter. I also find that there is some repetition of text in this chapter (Chapter 4.4) from previous part of the text.

11) Chapter 5 is also a result and should be part of Chapter 4. I do not think it is necessary to subdivide the sediment volume chapter into sub-chapters.

12) Chapter 6.1 is partly a review (e.g. lines 355-370) and partly a result/discussion text about how to calculate volumes (lines 383 - 400). Thus, part of the Chapter 6.1 text should be included in the result chapter and the entire Chapter 6.1 should instead includes a text about the development of the fjord system; telling the reader what have happened in the fjord during LGM and the last deglaciation (based on you data)

13) Chapter 6.2 also includes metodes (e.g. calculation of erosion rates in lines 435-450) which should be included in Chapter 3 and results (the erosion rates itself, e.g. lines 450-470) which should be in the result chapter. Chapter 6.2 should focus on comparison with other fjord systems (regarding erosion rates, volumes etc).

14) Some more references to fjord papers from UK, Norway and Svalbard should be implemented. The last few years numerous papers have been published, and which are relevant for the present manuscript.

---

## Editor Comment (EC1) · Chris R. Stokes (Editor) · 30 Sep 2019

I would like to thank both reviewers for their timely and constructive reviews of this manuscript. Both are generally positive, but make some helpful suggestions to improve the clarity and perhaps broaden the context and implications of the work. It remains my view that this is an important contribution and I would welcome the submission of a revised manuscript.
* * *

---

## Author Comment (AC1) · 1 Nov 2019

Below we list our responses to each of the comments and details of the changes made to the manuscript. Reviewers' comments are listed in sequence followed in each case by our responses.

Review: Anonymous Referee #1

This is a well written and illustrated paper that presents new data on glacigenic sediment fluxes and erosion rates associated with the retreat of Petermann Glacier in NW Greenland. The paper is very appropriate to The Cryosphere. However, there are a

few issues that need to be addressed:

1. You state that this is the first comprehensive investigation of the glacial-sedimentary infill of a major fjord system in Greenland (last para of Introduction and first line of Conclusions). I think you could do with justifying this claim a bit more robustly. For example the lead author has herself published on a similar topic from Disko Bugt/Jakobshavns Isbrae (e.g., Hogan et al., 2012 Marine Geology) and there has been previous work carried out on seismic stratigraphy in East Greenland fjords (Scoresby Sund – e.g., Uenzelmann-Neben, et al. 1991: Quaternary sediments in Scoresby Sund. East Greenland: their distribution from rellection seismic data. In Moller et al. (eds.): The Last Interglacial-Glacial Cycle: Jameson Land and Scoresby Sund, East Greenland). It may be that the present study significantly supersedes this earlier work but more justification/comment on this is required.

We thank the reviewer for this comment and have addressed it by adding text to several places in the text. We have now stated that the difference in this study is the combination of high-resolution acoustic methods with a high survey-line density INSIDE a major Greenland fjord and we have included references to previous geophysical studies of Greenland fjord infill: Uenzelmann-Neben et al., 1991; Andrews et al., 1994; Gilbert et al., 1998, 2002; Ó Cofaigh et al., 2001; Evans et al., 2002. An earlier draft of the paper actually included more information on the unique-ness of this high-resolution survey but that was cut at some stage!

2. Line 132. Explain why a delta-R of 268+/- 82 was chosen.

We have used delta-R of 268+/- 82 directly following the recalibration of terrestrial radiocarbon dates by Jakobsson et al. (2018), which we refer (and cite) to on lines 137-138 of the original manuscript. The reported 1-sigma ranges are those calculated by Jakobsson et al. (2018), available as this paper is from the same authorship group, and were included to give age ranges for the glacial erosion estimates. The delta-R of 268 +/- 82 used by Jakobsson et al. (2018) is based on the five nearest radiocarbondated shells to Nares Strait as reported in Coulthard et al. (2010), however, because the delta-R is derived from Jakobsson et al (2018) which is cited directly we do not think it necessary to add to the text to address this comment. Coulthard, R. D., Furze, M. F. A., Pieńkowski, A. J., Chantel Nixon, F. & England, J. H. New marine ΔR values for arctic Canada. Quat. Geochronol. 5, 419–434 (2010).

3. I suggest you change Sub-heading 4.2 'Petermann Fjord' to something a bit more informative, particularly as sub-heading 4.1 is 'Seismo-acoustic facies and depositional environments in Petermann Fjord and Nares Strait'.

We agree that the sub-headings could be more informative and have now changed section 4.2 heading to "Glacial marine sediment infill in Petermann Fjord and Nares Strait" (Sections 4.2 and 4.2 were merged in response to R2 comment #10). We believe that this new sub-heading describes the contents of these paragraphs more fully but does not wander into interpretation of the age of the sediments as post-glacial, Holocene or post-LGM, which would contradict comment #5 from this reviewer. All of the infilling sediments were deposited in a glaciomarine environment so we think this new sub-heading is appropriate.

4. Lines 230-233 and Line 316. You state that subglacial tills are deposited as sediment gravity flows. But gravity flows are NOT subglacial tills. For a recent treatment see D.J.A. Evans 2018 – 'Till'. Please change.

We see the confusion in the text regarding these statements and have now clarified it in both places by removing the term subglacial and adding a description of the depositional process (see tracked changes document). The confusion arises because subglacial material was deposited at the grounding line, and some of this subglacial material was remobilized in sediment gravity flows in front of the grounding line. Acoustically, there is no difference/boundary between these facies so they appear continuous in our profiles, however we recognise the different modes of deposition. We thank the reviewer for forcing us to look carefully at these statements!

5. Lines 369-371. You suggest that pre-LGM sediments most likely do not occur citing as support the presence of glacially-sculpted bedrock surfaces and referring to Jakobsson et al. (2018). I think this is pretty thin evidence. For example, pre-LGM sediments could be preserved locally within bedrock depressions and/or be too thin to be resolved by your seismic system(s). At the very least I would suggest the inclusion of a caveat acknowledging this would be appropriate here.

This comment is thought-provoking. We take the point that pre-LGM sediments can be preserved in bedrock depressions but we note that there are few large, deep basins filled with hundreds of meters of sediment in our study area (see Fig. 11a) as is the case with many of the cited examples that include pre-LGM sediments. Indeed, the bathymetry shows a sculpted seafloor for almost all of the fjord area which almost certainly was sculpted by ice flow during the last glacial (see Jakobsson et al., 2018). Furthermore, our calculations only include sediment that is above this "LGM surface" (which could be till or bedrock) meaning that, for the most part, our volume estimates should not include pre-LGM sediments. We do accept that any pre-LGM sediments could be too thin to be resolved in our SBP profiles and this is a potential source of error. However, if there are not many bedrock basins where pre-LGM sediments may be preserved, and/or pre-LGM sediments are thin then, volumetrically, they probably do not make up a large component of the infill and this assumption is appropriate for our study. However, as we acknowledge in the text, there is no way to distinguish these sediments from Holocene post-glacial sediments in our data and that there is no way to quantify what kind of error this might add to our estimates, which is unsatisfying. As such, we have added the caveat that pre-LGM sediments may be contributing to our volumes and that this is a potential source of error. Note that this text has been moved to the methods section as per R2 Comment #13.

6. Lines 372-373. You state that glaciomarine sedimentation seaward of the grounding line has two components the first of which is that of coarse or mixed material delivered to the grounding zone as subglacial deposits. But how is this material actually

deposited? It is glaciomarine sediment deposited seaward of the grounding line, so although it might be delivered to the grounding line subglacially it is not deposited seaward of the grounding line as a subglacial till.

We accept this comment and have modified the sentence to say that the material is delivered to the GL subglacially but deposited seaward of this as gravity flows (see tracked changes document).

---

## Author Comment (AC2) · 1 Nov 2019

Review: Anonymous Referee #2

The manuscript of Hogan et al. present new knowledge from the Petermann Fjord, and is based on seismic profiles. The study gives important information from a fjord environment that rarely are surveyed due to ice conditions and its remote location. The paper is suitable for publishing in the journal, but only after a moderate to major revision. The main comment to the paper is that it must be better structured and that the discussion has also to implement a depositional history of the fjord system.

[Figure]

Comments: 1) Chapter 1. Introduction: The paper should better explain why this study is of regional interest. As it is in the present paper the focus is quite local (Greenland). Why should we read this paper?

We have replaced the first paragraph of the Introduction to broaden the relevance of this paper and also adjusted the objectives to have a less regional focus.

2) Chapter 1. Introduction: In the listing of the objectives a more overall objective should also be included, e.g. compare erosion rates in this fjord environment with fjords elsewhere, global outlook, regional considerations. The objectives as they are now is quite "local".

We have now modified the objectives to be more regionally significant, see objective (4). They now read: "The objectives are: (1) to map glacial marine sediment units, interpret their seismic stratigraphy, and calculate their volumes; (2) to derive deglacial sediment fluxes and erosion rates; (3) to compare our results with other high-latitude fjord settings (Northern and Southern hemisphere) considering regional variations; and (4) to provide geological boundary conditions for numerical glacier modelling exercises."

3) Some more references to Figures 1 and 2 should be included in Chapter 1

We have now referred to these figures in Chapter 1.

4) In the heading of Chapter 2 delete "(geology, physiography, oceanography)"

This has been done.

5) Chapter 3 should be renamed to "Data and Methods" as you also describe the data used in the study

This has been done.

6) Chapter 3 does not include information about the cores you have used in the study. Information about these cores (and which now partly are in Supplementary Material) should be mentioned.

We appreciate this comment and we have no incorporated the seismo-acoustic/lithofacies correlation in Section 3.2. We have also added the figure (now Fig. 6) and table describing the cores to the main text.

7) All information provided in Supplementary Material should instead be implemented in the main text as I think the information in Supplementary Material is essential for the paper.

As described above, we have now added the core correlation material to the main text. We have chosen not to add the Supp. Fig. 2 (glacial catchment areas for erosion estimates) to the main text because the discussion around glacial erosion rates and how to define glacial catchment areas remains integrated in the results and discussion section (sections 4.5, 5.3) and, therefore, we elect to leave this as a supplementary figure.

8) The methods used in the papers (how to calculate drainage area, erosion rates etc) are as I read it spread a bit around in the paper. Please structure the paper in such a way that all your methods are included in Chapter 3.

We have now restructured the Methods section to include descriptions of how we calculate glacial sediment volumes, fluxes and erosion rates (section 3.4).

9) The Result chapter should be restructured. As it is now it is some repetition of the text. I suggest that you in 4.1 included information (text and figures) from the Supplementary Material in defining your facies. Furthermore, I suggest that Chapters 4.2 and 4.3 are merged to get a better overall view of the study area. You do not need to repeat the definition of the facies in chapters 4.2 and 4.3 (or what these facies are composed of) since you have already defined this in Chapter 4.1.

We have followed these suggestions and modified the Results chapter. 4.2 and 4.3 have been merged and unnecessary repetition of the acoustic facies has been removed. We have also incorporated the core lithofacies in to section 4.1.

10) Why do the manuscript have an own chapter on GZWs (Chapter 4.4). This chapter could also be merged into the merged 4.2-4.3 chapter. I also find that there is some repetition of text in this chapter (Chapter 4.4) from previous part of the text.

There is a separate section on GZWs because these landforms represent a discrete sedimentary deposit that has a different origin (subglacial deposition) than the adjoining sediments but must be accounted for when calculating glacial sediment fluxes and erosion rates. In addition, the GZWs have distinct significance for the interpretation of the glacial history and sedimentation processes that have produced (part of) the infill. Thus, we prefer to highlight them in a separate sub-section (4.3) but we have added a sentence at the start of this section to outline why these deposits are treated separately from the rest of the unlithified fill.

11) Chapter 5 is also a result and should be part of Chapter 4. I do not think it is necessary to subdivide the sediment volume chapter into sub-chapters.

We have now combined Chapter 5 with Chapter 4 with a new sub-section 4.5 being: "Unlithified sediment volumes". We have renumbered the following sections accordingly.

12) Chapter 6.1 is partly a review (e.g. lines 355-370) and partly a result/discussion text about how to calculate volumes (lines 383 - 400). Thus, part of the Chapter 6.1 text should be included in the result chapter and the entire Chapter 6.1 should instead includes a text about the development of the fjord system; telling the reader what have happened in the fjord during LGM and the last deglaciation (based on you data)

Following comment #11 Chapter 6 has become Chapter 5. We have now added a section describing the evolution of the infill as section 5.1 and we have moved the text describing how to calculate volumes to the methods chapter (section 3.4).

13) Chapter 6.2 also includes methods (e.g. calculation of erosion rates in lines 435-450) which should be included in Chapter 3 and results (the erosion rates itself, e.g.

lines 450-470) which should be in the result chapter. Chapter 6.2 should focus on comparison with other fjord systems (regarding erosion rates, volumes etc).

We have followed this advice and moved text describing methods to the methods chapter and created a new results chapter (4.5) to present the calculated fluxes and erosion rates.

14) Some more references to fjord papers from UK, Norway and Svalbard should be implemented. The last few years numerous papers have been published, and which are relevant for the present manuscript.

We have added the following more recent references from UK, Norwegian, and Svalbard fjords to the manuscript in Sections 3.2, 3.4, 5.2:

Bellwald,B., Hjelstuen, B.O., Sejrup, H.P., and Haflidason, H.: Postglacial mass movements and depositional environments in a high-latitude fjord system – Hardangerfjorden, Western Norway, Marine Geology, 379, 157-175, 2016.

Callard, S.L., Ó Cofaigh, C., Benetti, S., Chiverrell, R. C., Van Landeghem, K. J. J., Saher, M. H., Gales, J. A., Small, D., Clark, C. D., Livingstone, S. J., Fabel, D., and Moreton, S.G.: Extent and retreat history of the Barra Fan Ice Stream offshore western Scotland and northern Ireland during the last glaciation, Quaternary Science Reviews, 201, 280-302, 2018.

Nielsen, T. and Rasmussen, T. L.: Reconstruction of ice sheet retreat after the Last Glacial maximum in Storfjorden, southern Svalbard. Marine Geology 402, 228-243, 2018.

Stoker, M. S., Bradwell, T., Howe, J. A., Wilkinson, I. P., and McIntyre, K., Lateglacial ice-cap dynamics in NW Scotland: evidence from the fjords of the Summer Isles region, Quaternary Science Reviews, 28, 3161-3184, 2009.

A revised version of the manuscript with edits made in MS Word Track Changes is attached as a supplementary zip file.

Please also note the supplement to this comment:
https://www.the-cryosphere-discuss.net/tc-2019-171/tc-2019-171-AC2-supplement.zip
* * *

---

## Author Response (AR1)

**Author Response for: Glacial sedimentation, fluxes and erosion rates associated with ice retreat in Petermann Fjord and Nares Strait, NW Greenland**

Kelly A. Hogan[1,2], Martin Jakobsson[3,4], Larry Mayer[2], Brendan Reilly[5], Anne Jennings[6], Tove Nielsen[7], Katrine J. Andresen[8], Egon Nørmark[8], Katrien Heirman[7,10], Elina Kamla[7,9], Kevin Jerram[2], Christian Stranne[3,4], Alan Mix[5]

[1]British Antarctic Survey, Natural Environment Research Council, High Cross, Madingley Road, Cambridge, CB3 0ET, UK
[2] Center for Coast and Ocean Mapping, University of New Hampshire, NH 03824, USA
[3] Department of Geological Sciences, Stockholm University, 10691 Stockholm, Sweden
[4] Bolin Centre for Climate Research, Stockholm University, Stockholm 106 91, Sweden
[5] College of Earth, Ocean, and Atmospheric Sciences, Oregon State University, Corvallis, OR 97331, USA
[6] Institute of Arctic and Alpine Research, University of Colorado, Boulder, CO 80309-0450, USA
[7] Geological Survey of Denmark and Greenland, Øster Voldgade 10, 1350 Copenhagen K, Denmark
[8] Department of Geoscience, Aarhus University, Hoegh-Guldbergs Gade 2, DK-8000, Aarhus C, Denmark
[9] TNO, Geological Survey of the Netherlands, Princetonlaan 6, NL-3584 CB Utrecht, The Netherlands
[10] Rambøll Management Consulting, Hannemanns Allé 53, DK-2300 Copenhagen S, Denmark

*Correspondence to*: Kelly A. Hogan (kelgan@bas.ac.uk)

**Editor Decision: Publish subject to minor revisions (review by editor) (14 Nov 2019) by Chris R. Stokes**

**Comments to the Author:**

Thank you for your revised manuscript and your diligent response to the reviewer comments. Based on your response, and the revised version of the manuscript with tracked changes, I'm delighted to recommend publication. When you submit the final version of your revised manuscript, please carefully proof-read the manuscript. I noticed a small number of cases where the punctuation and grammar were awkward, mainly where new text had been added to sentences from the original manuscript (e.g. new text on lines 216-218).

**Author Response:** We thank the Editor for recommending our work for publication. We have now thoroughly re-read the manuscript and corrected awkward wording in several places including lines 216-218, 223-240, 382-387. We have also carefully checked for typos, corrected 10 errors in the reference list and several small typos in the Figures, mostly due to changes in the figure numbers during revision. On the next page, we attach a full version of the manuscript with tracked changes to show the edits made. Please note the author listing change with Alan Mix now appearing last in the list.

[revised manuscript text omitted]